# I Think, Therefore I Diffuse:
# Enabling Multimodal In-Context Reasoning in Diffusion Models

**Zhenxing Mi** [1,2]  **Kuan-Chieh Wang** [3]  **Guocheng Qian** [3]  **Hanrong Ye** [1]  **Runtao Liu** [1]
**Sergey Tulyakov** [3]  **Kfir Aberman** [3]  **Dan Xu** [1]

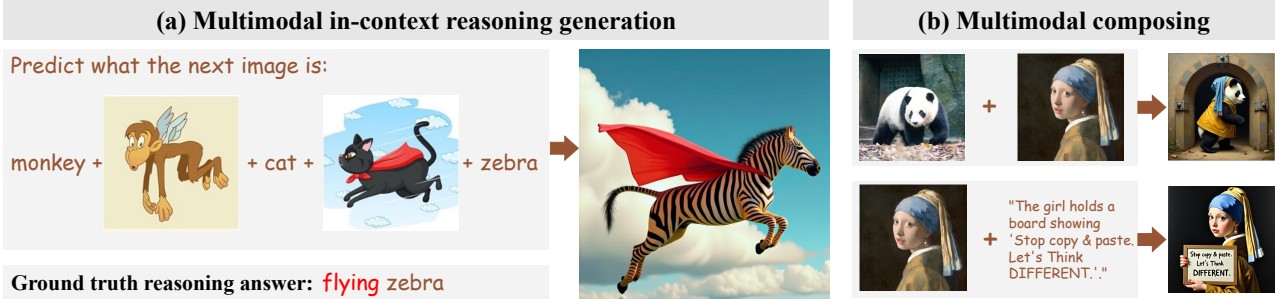

*Figure 1.* (a) Our ThinkDiff reasons over interleaved images (*a flying monkey* and *a flying cat*) and text prompts (*monkey*, *cat*, and *zebra*) to generate a logically correct and high-quality image (*a flying zebra*). The ground truth reasoning answer is provided as a reference for readers. (b) ThinkDiff composes images and texts into a coherent and reasonable image.

## Abstract

This paper presents ThinkDiff, a novel alignment paradigm that empowers text-to-image diffusion models with multimodal in-context understanding and reasoning capabilities by integrating the strengths of vision-language models (VLMs). Existing multimodal diffusion finetuning methods largely focus on pixel-level reconstruction rather than in-context reasoning, and are constrained by the complexity and limited availability of reasoning-based datasets. ThinkDiff addresses these challenges by leveraging vision-language training as a proxy task, aligning VLMs with the decoder of an encoder-decoder large language model (LLM) instead of a diffusion decoder. This proxy task builds on the observation that the **LLM decoder** shares the same input feature space with **diffusion decoders** that use the corresponding **LLM encoder** for prompt embedding. As a result, aligning VLMs with diffusion decoders can be simplified through alignment with the LLM de-

coder. Without complex training and datasets, ThinkDiff effectively unleashes understanding, reasoning, and composing capabilities in diffusion models. Experiments demonstrate that ThinkDiff significantly improves accuracy from 19.2% to 46.3% on the challenging CoBSAT benchmark for multimodal in-context reasoning generation, with only 5 hours of training on 4 A100 GPUs. Additionally, ThinkDiff demonstrates exceptional performance in composing multiple images and texts into logically coherent images. Project page: *https://mizhenxing.github.io/ThinkDiff* .

## 1. Introduction

*Can diffusion models take "IQ tests"?* Figure 1a presents an example of a visual analogy IQ test. The model is provided with images of *a flying monkey* and *a flying cat*, along with text prompts of *monkey*, *cat*, and *zebra*, and asked to generate the next image. A reasonable output image should be an image of *a flying zebra*, requiring the model's ability to reason and recognize implicit patterns in context, such as the shared attribute of the *flying* action in this example.

The concept of enabling diffusion models to think and then generate is compelling yet underexplored. Current text-to-image diffusion models (AI, 2024c; Forest, 2024a) excel at generating high-quality images by strictly following explicit prompts, while typically lacking multimodal in-context rea-

---

[1]Department of Computer Science and Engineering (CSE), The Hong Kong University of Science and Technology (HKUST). [2]This work was partially done while interning at Snap. [3]Snap Inc. Correspondence to: Zhenxing Mi <zmiaa@connect.ust.hk>.

*Proceedings of the 42$^{nd}$ International Conference on Machine Learning*, Vancouver, Canada. PMLR 267, 2025. Copyright 2025 by the author(s).

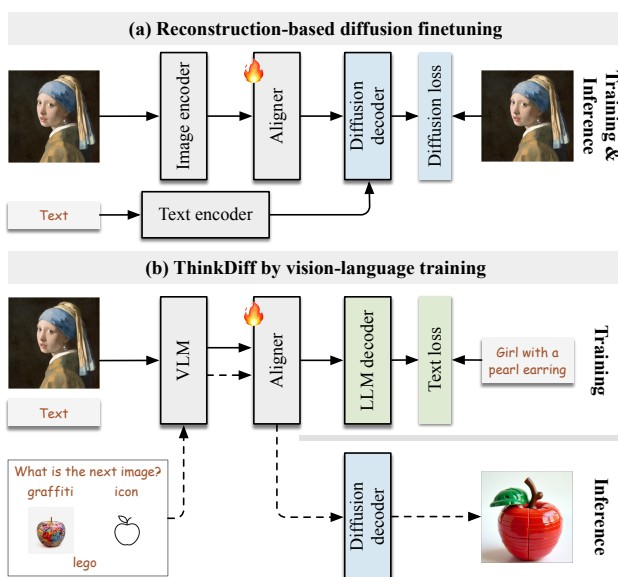

*Figure 2.* **(a)** Reconstruction-based diffusion finetuning integrates image features using a diffusion loss, focusing on pixel-level image reconstruction without reasoning. **(b)** ThinkDiff aligns a VLM to an LLM decoder by vision-language training on image-caption datasets. In inference (dotted lines), it transfers multimodal in-context reasoning capabilities from the VLM to a diffusion decoder.

soning. Unlocking reasoning capabilities in them can enable them to handle more sophisticated tasks, such as interpreting complex instructions, solving visual analogy problems that require inferring implicit logic relationships, and composing multiple images and text in a logically consistent manner.

With rapid advancements in vision-language models (VLMs) such as CLIP (Radford et al., 2021) and GPT-like models (Radford et al., 2018), we now have powerful tools for advanced multimodal understanding and reasoning. This leads us to a question: *can we equip diffusion models with the reasoning capabilities of VLMs?*

Existing multimodal diffusion adapters (Zhang et al., 2023; Ye et al., 2023; Mou et al., 2024) primarily rely on reconstruction-based diffusion finetuning to incorporate visual conditions into text-to-image diffusion models. Figure 2a illustrates the typical training pipeline of IP-Adapter (Ye et al., 2023), where the model is finetuned to replicate input images at the pixel level. While effective for pixel-level control and high-fidelity image generation, adapting this finetuning paradigm to support in-context reasoning introduces several challenges. **First**, this multimodal finetuning primarily focuses on pixel-level reconstruction of explicit image inputs rather than performing multimodal reasoning based on input context. **Second**, the pixel-level reconstruction training does not focus on aligning vision representations with the textual feature space, limiting the model's ability to reason effectively across modalities.

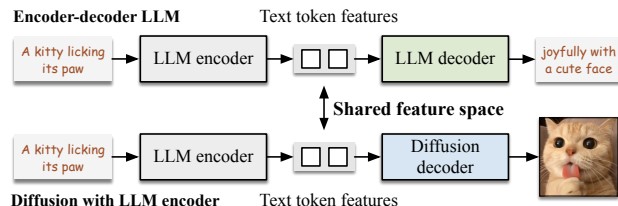

*Figure 3.* Several diffusion models share a language encoder with encoder-decoder LLMs, allowing aligning with diffusion decoders through aligning with LLM decoders.

**Third**, instead of readily available image-caption pairs, it requires multimodal reasoning datasets that pair multimodal inputs with logically consistent output images and cover different reasoning tasks. Collecting such datasets is significantly more complex than captioning images. Existing instruction-guided datasets such as the synthetic Instruct-Pix2Pix (Brooks et al., 2023) dataset primarily focus on image editing tasks, lacking the diversity needed for reasoning-based generation tasks. **Finally**, finetuning diffusion models for reasoning from scratch using limited datasets constrains their performance across a broad range of reasoning tasks.

To tackle these challenges, we propose **ThinkDiff**, a novel alignment paradigm to transfer multimodal in-context reasoning capabilities from VLMs to diffusion models. Instead of directly aligning VLMs with a diffusion decoder, we design a proxy task to align VLMs with a large language model (LLM) decoder by vision-language training. The foundation of this proxy task is depicted in Figure 3. Recent diffusion models (AI, 2024b; Chen et al., 2024; Forest, 2024a; AI, 2024c) have adopted the **encoder** of an encoder-decoder LLM (Raffel et al., 2020) as diffusion models' prompt encoder. This shared text encoder establishes a shared input feature space for both the diffusion **decoder** and LLM **decoder**. Therefore, aligning a VLM with a diffusion decoder can be achieved by the proxy task of aligning a VLM with the LLM decoder by vision-language training.

Figure 2b depicts the vision-language training in ThinkDiff. The input images and text prompts are processed by a VLM and an aligner network, after which they are fed into an LLM decoder. The LLM decoder generates text autoregressively, supervised by a cross-entropy loss against ground truth texts. After training, the VLM is aligned to the LLM decoder, and inherently to the diffusion decoder.

Our method offers several advantages. **First**, it fully leverages the multimodal in-context understanding and reasoning capabilities of VLMs without requiring expensive training from scratch. **Second**, by aligning multimodal features to the input space of the LLM decoder through fine-grained text supervision, the model effectively captures rich semantic details from multimodal inputs, enabling seamless collaboration between vision and text modalities. **Finally**, ThinkDiff is lightweight, efficient and highly versatile. The

vision-language training in it only requires readily available image-caption pairs, eliminating the need for complex reasoning-based datasets while achieving remarkable in-context reasoning capabilities.

This paper introduces two variants of ThinkDiff, each using a different VLM. ThinkDiff-LVLM aligns generated tokens of a large vision-language model (LVLM) to diffusion models. ThinkDiff-CLIP aligns image tokens from a CLIP vision encoder (Radford et al., 2021) to diffusion models. Our contributions are summarized as follows:

- We propose ThinkDiff, a novel alignment paradigm that equips diffusion models with multimodal in-context reasoning capabilities from VLMs.
- ThinkDiff designs a proxy task to align VLMs into a shared feature space of both an LLM decoder and a diffusion decoder by vision-language training, fully transferring VLM's reasoning capabilities to diffusion models with efficient training and simple datasets.
- We address the poor convergence problem in ThinkDiff for robust feature alignment. After training for only 5 hours on 4 A100 GPUs, ThinkDiff improves state-of-the-art accuracy on the major visual in-context learning benchmark (Zeng et al., 2024) from 19.2% to 46.3%. It also demonstrates powerful abilities to compose multiple images and texts into logically coherent images.

## 2. Related Work

### 2.1. Diffusion Models

Diffusion models have become powerful tools for text-to-image generation (Ho et al., 2020; Rombach et al., 2022; Forest, 2024a). Early models, e.g. Stable Diffusion (Rombach et al., 2022), use CLIP (Radford et al., 2021) for prompt embedding, while recent works integrate large language models (LLMs) (Saharia et al., 2022; Chen et al., 2024; AI, 2024c) for complex prompts. Methods such as ControlNet (Zhang et al., 2023), T2I-Adapter (Mou et al., 2024), and IP-Adapter (Ye et al., 2023) introduce structural and image-level controls by reconstruction-based fine-tuning. Personalized generation has been enhanced by methods like DreamBooth (Ruiz et al., 2023), and other methods (Gal et al., 2023; Wang et al., 2024a; Li et al., 2024; Wang et al., 2024c; Qian et al., 2024; Wang et al., 2024d), some of which use interleaved image-text inputs (Pan et al., 2023; Berman & Peysakhovich, 2024). However, these methods focus on reconstruction fidelity rather than in-context reasoning. In contrast, our method equips diffusion models with the multimodal in-context reasoning capabilities of VLMs.

### 2.2. Unified Understanding and Generation

Recent work on large language models (LLMs) and diffusion transformers (Peebles & Xie, 2023; Forest, 2024a) has inspired unified models for multimodal understanding and generation. These models either finetune LLMs to generate image tokens, which are then decoded into images via diffusion decoders (Ge et al., 2024; Pan et al., 2023; Sun et al., 2023; Koh et al., 2024; Wu et al., 2023; Ye et al., 2024), or integrate text, image, and noise tokens within a transformer architecture (Xiao et al., 2024; Shi et al., 2024). They are typically trained end-to-end with diffusion losses or align output image tokens with CLIP text features using cosine similarity losses (Wu et al., 2023; Ye et al., 2024; Tong et al., 2024). While some methods exhibit preliminary reasoning capabilities, these capabilities remain constrained by the limits of diffusion training paradigms, the availability of reasoning datasets, and the representational limits of CLIP embeddings. In contrast, our method leverages vision-language training to transfer advanced multimodal reasoning capabilities in VLMs to diffusion models.

### 2.3. Vision-language Training

Vision-language training has proven effective in developing powerful multimodal models. CLIP-like models (Radford et al., 2021; Fang et al., 2023; Girdhar et al., 2023) use contrastive learning to align image and text embeddings. Recent large vision-language models (LVLMs)(Li et al., 2023; Liu et al., 2023; Zhu et al., 2023; AI, 2024a; Wang et al., 2024b) align CLIP visual features with advanced large language models (LLMs)(Brown et al., 2020; Achiam et al., 2024; AI, 2024a; Yang et al., 2024a) by fine-grained text prediction. This vision-language training enables robust multimodal feature alignment, developing multimodal understanding and reasoning by leveraging powerful LLMs. Inspired by these advancements, our method employs vision-language training as a proxy task to bridge VLMs with diffusion models, inheriting their advanced multimodal reasoning capabilities.

## 3. Method

### 3.1. Overview

ThinkDiff employs VLMs to enable diffusion decoders to perform multimodal in-context reasoning. This is achieved by an aligner network that bridges a VLM and a diffusion decoder. As described in Section 1, ThinkDiff simplifies the alignment process by introducing a proxy task that aligns the VLM with an LLM decoder using text supervision. This task is based on the shared input feature space between the LLM decoder and diffusion decoder. Figure 2b and Figure 4 illustrate the overall network structure and two model variants, respectively. The multimodal input comprises a set of images $\{I_i\}$ and text tokens $\{T_i\}$. The aligner network processes its input token features $\{x_i\}$ into its output token features $\{x_i'\}$. In training, ThinkDiff generates text tokens $\{y_i'\}$, supervised by ground truth text tokens $\{y_i\}$. In inference, it generates an image $I'$.

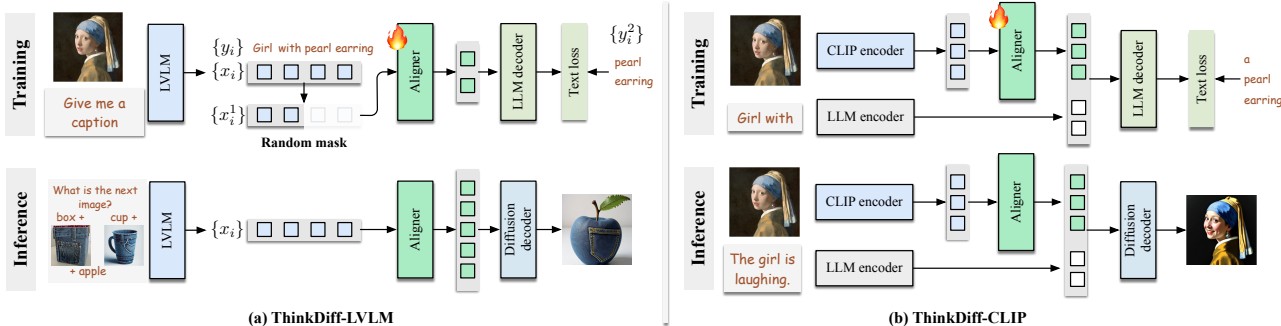

*Figure 4.* (a) In **ThinkDiff-LVLM** training, the LVLM processes an image and a text to generate text tokens and token features, with some token features randomly masked. Unmasked token features are passed to a trainable aligner network and an LLM decoder, predicting masked text tokens supervised by cross-entropy loss. In inference, the LLM decoder is replaced by a diffusion decoder, enabling in-context reasoning image generation from interleaved images and texts. (b) In **ThinkDiff-CLIP** training, a CLIP vision model extracts image token features which are then mapped by a trainable aligner network. A part of the image caption is encoded by the LLM encoder and concatenated with image tokens. These combined tokens are passed to the LLM decoder to predict the next part of the caption supervised by cross-entropy loss. In inference, the LLM decoder is replaced by a diffusion encoder, allowing coherent image generation based on multimodal context.

**Module Overview.** ThinkDiff comprises three submodules: a source VLM ($\mathcal{M}_{\text{VLM}}$), an aligner network ($\mathcal{M}_{\text{AN}}$), and a decoder. The decoder is a LLM decoder ($\mathcal{M}_{\text{LLMD}}$) in training and a diffusion decoder ($\mathcal{M}_{\text{DiffD}}$) in inference.

**Source VLM.** The source VLM generates multimodal token features $\{x_i\}$, capturing the reasoning and understanding derived from multimodal inputs and transferring this information to diffusion decoders. The generation is expressed as: $\{x_i\} = \mathcal{M}_{\text{VLM}}(\{I_i\}, \{T_i\})$. This paper introduces two variants of ThinkDiff, each utilizing a different VLM. ThinkDiff-LVLM uses a large vision-language model (LVLM) to deliver advanced multimodal reasoning capabilities while ThinkDiff-CLIP leverages the semantically rich image embeddings provided by a CLIP vision encoder for image understanding. Detailed descriptions of these variants can be found in Sections 3.3 and 3.4.

**Aligner network.** The aligner network bridges the source VLM with the LLM and diffusion decoder. It transforms token features $\{x_i\}$, which encapsulate rich reasoning information, into $\{x'_i\}$, making them interpretable by the LLM and diffusion decoder. This transformation is represented as: $\{x'_i\} = \mathcal{M}_{\text{AN}}(\{x_i\})$.

**Decoder.** The decoder operates differently during training and inference. The LLM decoder ($\mathcal{M}_{\text{LLMD}}$) is central to ThinkDiff's vision-language training. It is derived from an encoder-decoder LLM. In this LLM, the LLM encoder encodes token features and the LLM decoder generates text autoregressively from these token features. In ThinkDiff training, the VLM token features $\{x_i\}$ are mapped to $\{x'_i\}$ by the aligner network. The LLM decoder then treats $\{x'_i\}$ as if they were outputs from the LLM encoder and autoregressively decodes them into text $\{y'_i\}$. This process

is expressed as: $\{y'_i\} = \mathcal{M}_{\text{LLMD}}(\{x'_i\})$. By this training, VLM token features are aligned with the decoder's input space, transferring reasoning capabilities from the VLM to $\mathcal{M}_{\text{LLMD}}$ in training and to $\mathcal{M}_{\text{DiffD}}$ in inference.

In inference, the LLM decoder is replaced by a diffusion decoder ($\mathcal{M}_{\text{DiffD}}$), which can interpret VLM's outputs and leverage the VLM's multimodal reasoning abilities for image generation. ThinkDiff can handle multiple images, texts, or interleaved sequences of images and texts during inference, thanks to their shared feature space. The generated image $I'$ is given by $I' = \mathcal{M}_{\text{DiffD}}(\{x'_i\})$.

**Loss.** We employ a cross-entropy loss between the LLM decoder's generated tokens $\{y'_i\}$ and the ground truth text tokens $\{y_i\}$ in training. Let $N$ be the length of $\{y'_i\}$, the loss is defined as: $L_{\text{text}} = -\frac{1}{N}\sum_{i=1}^{N} \log p(y'_i = y_i)$.

In the following sections, we detail the design of the aligner network and two variants of ThinkDiff.

### 3.2. Aligner Network

The aligner network $\mathcal{M}_{\text{AN}}$ is a lightweight module comprising two linear layers ($\mathcal{L}_{\text{Linear}}$), a GELU activation ($\mathcal{L}_{\text{GELU}}$) and an RMSNorm layer (Zhang & Sennrich, 2019) ($\mathcal{L}_{\text{Norm}}$). Given the VLM's output $\{x_i\}$, the output $\{x'_i\}$ of $\mathcal{M}_{\text{AN}}$ is:

$$\{x'_i\} = \mathcal{L}_{\text{Norm}}(\mathcal{L}_{\text{Linear}}(\mathcal{L}_{\text{GELU}}(\mathcal{L}_{\text{Linear}}(\{x_i\})))) \quad (1)$$

In training, only $\mathcal{M}_{\text{AN}}$ is updated. Despite its simplicity, $\mathcal{M}_{\text{AN}}$ can effectively aligns feature spaces of the powerful VLM and the LLM decoder in the training.

**Stable training.** Our experiments revealed that without a carefully initialized RMSNorm layer, ThinkDiff encounters

convergence issues due to a scale mismatch between the VLM output space and the LLM decoder input space. To address this, we incorporate an RMSNorm (Zhang & Sennrich, 2019) layer into $\mathcal{M}_{\text{AN}}$, initialized with parameters from the LLM **encoder's** final RMSNorm layer. Since the LLM encoder output space aligns naturally with the LLM decoder input space, this initialization ensures consistent scale alignment at the start of training, significantly improving training stability and convergence.

### 3.3. ThinkDiff-LVLM

ThinkDiff-LVLM incorporates a decoder-only large vision-language model (LVLM) that excels at advanced in-context reasoning tasks, as its VLM. It aligns the deep features of the LVLM's **generated** tokens to both $\mathcal{M}_{\text{LLMD}}$ and $\mathcal{M}_{\text{DiffD}}$.

**Training.** The training framework is illustrated in Figure 4a. The LVLM autoregressively generates text tokens $\{y_i\}$ from an input image $I$ and text prompt $T$. The corresponding token features $\{x_i\}$ are extracted from the LVLM's final RMSNorm layer. These features $\{x_i\}$ are then passed to $\mathcal{M}_{\text{AN}}$ and $\mathcal{M}_{\text{LLMD}}$, where they are decoded into text tokens $\{y_i'\}$, supervised by LVLM's generated tokens $\{y_i\}$. This setup is self-supervised, as both the token features $\{x_i\}$ and the supervision $\{y_i\}$ are all generated by the LVLM itself. This enables the aligner network to accurately transfer information from the LVLM to $\mathcal{M}_{\text{LLMD}}$ and $\mathcal{M}_{\text{DiffD}}$.

However, in this setup, token features $\{x_i\}$ have a one-to-one correspondence with the supervision text tokens $\{y_i\}$. This may cause the aligner to learn a trivial mapping between $\{x_i\}$ and $\{y_i\}$ without truly aligning features. We refer to this issue as "shortcut mapping".

**Random masked training.** To address the "shortcut mapping" issue, we introduce a random masked training strategy. In this strategy, text tokens $\{y_i\}$ and features $\{x_i\}$ are randomly split into two parts: $\{y_i^1\}$, $\{y_i^2\}$ and $\{x_i^1\}$, $\{x_i^2\}$, where $\{y_i^1\}$ correspond to $\{x_i^1\}$ and $\{y_i^2\}$ correspond to $\{x_i^2\}$. Only the first part $\{x_i^1\}$ is passed to the aligner and LLM decoder, generating text tokens $\{y_i'\}$ supervised by the second part of tokens $\{y_i^2\}$. This breaks the one-to-one correspondence, encouraging a more robust feature alignment. The generated tokens $\{y_i'\}$ are computed as:

$$\{y_i'\} = \mathcal{M}_{\text{LLMD}}(\mathcal{M}_{\text{AN}}(f_{\text{mask}}(\mathcal{M}_{\text{LVLMG}}(I, T)))), \quad (2)$$

where $f_{mask}$ is the random masking and $\mathcal{M}_{\text{LVLMG}}$ is the LVLM's generation process. The cross-entropy loss is: $L_{\text{LVLM}} = -\frac{1}{N}\sum_{i=1}^{N} \log p(y_i' = y_i^2)$.

**Why use LVLM's generated tokens.** Some diffusion models (Liu et al., 2024; Xie et al., 2024) incorporate decoder-only LLMs for prompt encoding but actually treat them as **encoders** by using the deep features of **input** tokens. In contrast, ThinkDiff-LVLM uses the deep features of the

**generated** tokens from the LVLM decoder as input to the aligner. This design is motivated by the insight that, in autoregressive models, reasoning is embedded in the generation process. Tokens are generated sequentially, conditioned on both the input context and the prior generated tokens. As a result, the full sequence of generated tokens captures the model's logical reasoning about the input context. By aligning these generated token features with diffusion models, ThinkDiff-LVLM ensures that the diffusion models inherit the LVLM's advanced multimodal reasoning capabilities.

**Inference for in-context reasoning.** In inference, as shown in Figure 4a, the LLM decoder is replaced by a diffusion decoder for image generation. As shown in Figure 1a and 5, ThinkDiff-LVLM effectively leverages the LVLM's multimodal in-context reasoning capability, using the context of interleaved images $\{I_i\}$ and texts $\{T_i\}$ to generate high-quality, logically coherent images that go beyond simply reconstructing the input content. The generated image $I'$ is:

$$I' = \mathcal{M}_{\text{DiffD}}(\mathcal{M}_{\text{AN}}(\mathcal{M}_{\text{LVLMG}}(\{I_i\}, \{T_i\}))) \quad (3)$$

### 3.4. ThinkDiff-CLIP

ThinkDiff-CLIP employs the vision encoder of a CLIP vision-language model (Radford et al., 2021) pretrained on contrastive vision-language tasks, as its VLM. This encoder produces semantically rich image features, enabling aligned diffusion decoders to generate images based on the semantic understanding of input images.

**Training.** Figure 4b illustrates the training framework. The model is trained to predict partial captions for an input image. The CLIP vision encoder encodes the input image $I$ into image tokens $\{x_i\}$, which are downsampled via 2D pooling to reduce token count. The aligner network then maps $\{x_i\}$ to $\{x_i'\}$. Meanwhile, the image caption $T$ is randomly split into two parts: $T_1$ and $T_2$. The first part, $T_1$, is encoded into text token features $\{t_i\}$ by the LLM encoder. The aligned image tokens $\{x_i'\}$ are concatenated with $\{t_i\}$, and fed to the LLM decoder, which autoregressively predicts text $\{y_i'\}$ supervised by the second caption part $T_2$ (tokens $\{y_i^2\}$). The text generation process is formulated as:

$$\{y_i'\} = \mathcal{M}_{\text{LLMD}}(f_{\text{cat}}(\mathcal{M}_{\text{AN}}(\mathcal{M}_{\text{CLIP}}(I)), \mathcal{M}_{\text{LLME}}(T_1))), \quad (4)$$

where $f_{\text{cat}}$ denotes concatenation, and $\mathcal{M}_{\text{LLME}}$ is the LLM encoder. The cross-entropy loss is: $L_{\text{CLIP}} = -\frac{1}{N}\sum_{i=1}^{N} \log p(y_i' = y_i^2)$. After training, the aligned image tokens $\{x_i'\}$ capture semantic details of the input image and can be interpreted by both $\mathcal{M}_{\text{LLMD}}$ and $\mathcal{M}_{\text{DiffD}}$.

**Inference.** In inference, as shown in Figure 4b, the LLM decoder is replaced by a diffusion decoder for image generation. As shown in Figure 1b, 6, 8, and 13, with an image as input, ThinkDiff-CLIP preserves semantic details of this

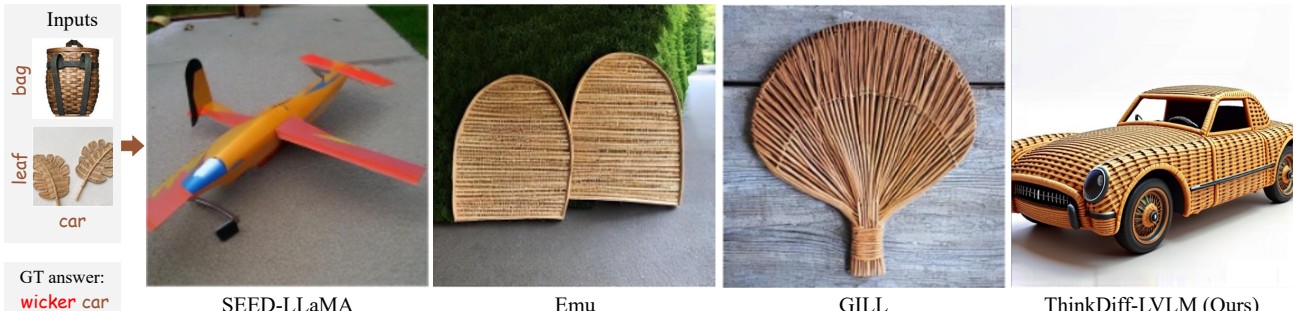

*Figure 5.* 2-shot evaluation results on CoBSAT. The input structure is similar to Figure 1a. Given multimodal inputs, ThinkDiff-LVLM accurately captures both implicit attributes (e.g., wicker material) and explicit attributes (e.g. car), and generates a logically correct image (wicker car). In contrast, methods such as SEED-LLaMA (Ge et al., 2024), Emu (Sun et al., 2023) and GILL (Koh et al., 2024) produce inaccurate and lower-quality images. The ground truth implicit attribute is highlighted in red for readers' reference. See more results in Appendix Figure 9 and 10.

*Table 1.* 2-shot CoBSAT accuracy of ThinkDiff-LVLM. It achieves SoTA accuracy on 9 of 10 tasks by large margins, increasing accuracy by more than 20% on Action-I, Color-II, Action-II tasks which are particularly hard for other methods.

| | Color-I | Background-I | Style-I | Action-I | Texture-I | Color-II | Background-II | Style-II | Action-II | Texture-II |
|---|---|---|---|---|---|---|---|---|---|---|
| SEED-LLaMA | **0.680** | 0.348 | 0.203 | 0.182 | 0.196 | 0.287 | 0.467 | 0.297 | 0.261 | 0.163 |
| Emu | 0.065 | 0.051 | 0.057 | 0.052 | 0.078 | 0.062 | 0.109 | 0.081 | 0.092 | 0.074 |
| GILL | 0.171 | 0.054 | 0.069 | 0.063 | 0.074 | 0.010 | 0.043 | 0.024 | 0.022 | 0.040 |
| ThinkDiff-LVLM | 0.622 | **0.349** | **0.237** | **0.459** | **0.290** | **0.511** | **0.534** | **0.340** | **0.534** | **0.292** |

image in the generated image. With multiple input images and text prompts, it seamlessly combines them into a semantically coherent image, as both image and text features are well-aligned within a shared feature space. These results highlight ThinkDiff-CLIP's ability to understand and compose multimodal context. In contrast, reconstruction-based diffusion finetuning methods like FLUX Ultra (Forest, 2024a), often struggle to simultaneously adhere to image and text prompts. The generation of ThinkDiff-CLIP is:

$$I' = \mathcal{M}_{\text{DiffD}}(f_{\text{cat}}(\mathcal{M}_{\text{AN}}(\mathcal{M}_{\text{CLIP}}(\{I_i\})), \mathcal{M}_{\text{LLME}}(\{T_i\})))$$

(5)

## 4. Experiments

### 4.1. Implement Details

**Base models.** We use publicly available FLUX.1-dev (Forest, 2024a) as the diffusion decoder as it employs T5 (Raffel et al., 2020), an LLM, as its prompt encoder. We use the corresponding T5 decoder as $\mathcal{M}_{\text{LLMD}}$. ThinkDiff-LVLM uses Qwen2-VL (Wang et al., 2024b) as the VLM, which excels at vision-language reasoning on interleaved images and texts. ThinkDiff-CLIP employs the vision encoder from the ViT-G/14 model of EVA-CLIP (Fang et al., 2023).

**Training and evaluation resources.** We use public image-caption datasets for training. ThinkDiff-LVLM is trained for 25,000 steps on 4 A100 GPUs for 5 hours, with a total batch size of 96. ThinkDiff-CLIP is trained for 100,000 steps on 4 A100 GPUs by one day, with a total batch size of 168. See Appendix B for detailed dataset settings. The multimodal

in-context reasoning capabilities of ThinkDiff-LVLM are evaluated on the challenging CoBSAT benchmark (Zeng et al., 2024) and measured by prediction accuracy. More details are in its paper. We assess ThinkDiff-CLIP's reasoning and composition abilities on various prompts and images from (Ruiz et al., 2023; Peng et al., 2024; Ye et al., 2023).

**Baselines.** We compare ThinkDiff-LVLM with SEED-LLaMA (Ge et al., 2024), Emu (Sun et al., 2023) and GILL (Koh et al., 2024) that can generate images based on image and text inputs. SEED-LLaMA is the previous state-of-the-art (SoTA) model on the CoBSAT benchmark. We compare ThinkDiff-CLIP with FLUX1.1-pro-Ultra API (Forest, 2024b), which supports image generation from image and text inputs. FLUX1.1-pro-Ultra is possibly finetuned by diffusion training and image reconstruction supervision, which differs fundamentally from our method.

### 4.2. Evaluation Results of ThinkDiff-LVLM

We evaluate ThinkDiff-LVLM on the 10 multimodal in-context reasoning generation tasks in the CoBSAT, in both 2-shot and 4-shot settings. In each setting, 2 or 4 input images and corresponding texts are provided as input, with an additional instruction prompt to make the model generate the next image that contains the correct object and attribute, based on in-context reasoning, (see Appendix Section B). Tables 1 and 2 report the accuracy for 2-shot and 4-shot evaluations, respectively. Results of SEED-LLaMA (Ge et al., 2024), Emu (Sun et al., 2023) and GILL (Koh et al., 2024) are token from the CoBSAT (Zeng et al., 2024) paper.

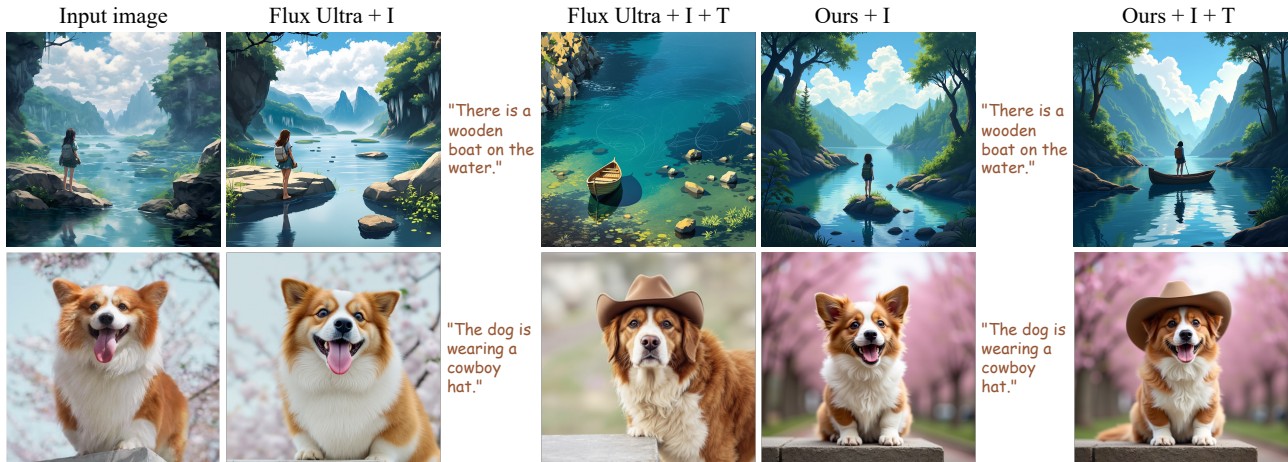

| Input image | Flux Ultra + I | Flux Ultra + I + T | Ours + I | Ours + I + T |
|---|---|---|---|---|

*Figure 6.* Generation results for single image (I) and single image with text prompt (I + T) inputs. Our method effectively integrates semantic details of both image and text modalities to produce coherent images. FLUX excels at replicating the input image but struggles to maintain consistency with additional text prompts. See more results in Figure 11.

*Table 2.* 4-shot CoBSAT accuracy of ThinkDiff-LVLM shows a 27% average improvement over other methods and a 4.7% increase over its 2-shot results, highlighting its ability to handle complex in-context reasoning. In contrast, SEED-LLaMA (Ge et al., 2024), Emu (Sun et al., 2023), and GILL (Koh et al., 2024) exhibit reduced performance in 4-shot evaluations, indicating their struggle with increased input complexity. Improvement ratios over SoTA are also provided.

| | Color-I | Background-I | Style-I | Action-I | Texture-I | Color-II | Background-II | Style-II | Action-II | Texture-II |
|---|---|---|---|---|---|---|---|---|---|---|
| SEED-LLaMA | 0.482 | 0.211 | 0.141 | 0.053 | 0.122 | 0.252 | 0.076 | 0.268 | 0.207 | 0.105 |
| Emu | 0.063 | 0.018 | 0.045 | 0.048 | 0.097 | 0.037 | 0.122 | 0.109 | 0.077 | 0.088 |
| GILL | 0.106 | 0.044 | 0.041 | 0.073 | 0.087 | 0.022 | 0.059 | 0.044 | 0.032 | 0.067 |
| Ours | **0.638** | **0.362** | **0.254** | **0.434** | **0.317** | **0.610** | **0.590** | **0.432** | **0.664** | **0.332** |
| **Improvement ($\Delta$%)** | **32.4%** | **71.6%** | **80.1%** | **718.9%** | **159.8%** | **142.1%** | **676.3%** | **61.2%** | **220.8%** | **216.2%** |

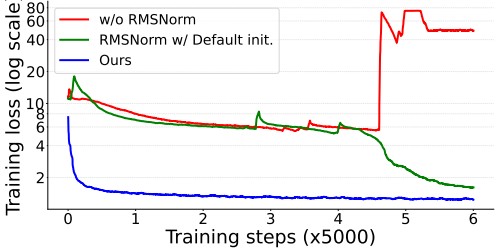

*Figure 7.* Training losses (log scale) of ThinkDiff-LVLM comparing different RMSNorm designs. Disabling RM-SNorm (w/o RMSNorm) or using the default RMSNorm initialization (RMSNorm w/ Default init.) results in significantly unstable training.

As shown in Table 1 for 2-shot evaluation, ThinkDiff-LVLM achieves SoTA performance on 9 out of 10 tasks, outperforming other methods by a large margin. Baselines like Emu and GILL perform poorly on most tasks with accuracy below 10%, reflecting the difficulty of these tasks. While SEED-LLaMA performs well on task Color-I, it underperforms ThinkDiff-LVLM on other tasks. Notably, ThinkDiff-LVLM exceeds the previous SoTA by over 20% in accuracy on Action-I, Color-II, and Action-II tasks, showcasing its superior in-context reasoning generation capabilities.

More importantly, in the more complex 4-shot evaluation (Table 2), ThinkDiff-LVLM further demonstrates its superior performance, outperforming all methods across every task, with an average accuracy improvement of 27%. Notably, it also shows a consistent 4.7% accuracy increase over its 2-shot performance, highlighting its ability to effectively leverage additional complex information. In contrast, the accuracy of baselines drops significantly with 4-shot inputs, indicating their difficulties with the increased complexity of multimodal inputs. This underscores that ThinkDiff-LVLM not only excels in advanced in-context reasoning but also adapts more effectively to complex multimodal inputs. Figures 5, 9, and 10 present the qualitative comparison, where ThinkDiff-LVLM generates both correct and significantly higher-quality images compared to other methods.

**Generation quality.** We further evaluate the quality of general image-conditioned generation of ThinkDiff-LVLM on 1k images in COCO (Lin et al., 2014). The model is conditioned by an image in the experiment. We show the FID, the CLIP image metric (CLIP-I), and the CLIP-Score (CLIP-T) (Hessel et al., 2021) in the Table 4. Our method can achieve much better performance than existing competitors such as SEED-LLaMA, Emu, and GILL. We also evaluate the general text-conditioned generation

*Table 3.* 2-shot results on CoBSAT ablating models with and without masking, and using deep features of input tokens.

|  | Color-I | Background-I | Style-I | Action-I | Texture-I | Color-II | Background-II | Style-II | Action-II | Texture-II |
|---|---|---|---|---|---|---|---|---|---|---|
| Ours using input tokens | 0.024 | 0.004 | 0.03 | 0.011 | 0.032 | 0.007 | 0.008 | 0.012 | 0.019 | 0.011 |
| Ours w/o masked training | 0.548 | 0.215 | 0.105 | 0.256 | 0.187 | 0.510 | 0.338 | 0.156 | 0.325 | 0.228 |
| Ours | **0.622** | **0.349** | **0.237** | **0.459** | **0.290** | **0.511** | **0.534** | **0.340** | **0.534** | **0.292** |

*Table 4.* Image-conditioned generation quality of ThinkDiff-LVLM on CLIP metrics on COCO.

| Model | CLIP-I↑ | CLIP-T↑ | FID↓ |
|---|---|---|---|
| SEED-LLaMA | 0.695 | 0.546 | 71.7 |
| Emu | 0.443 | 0.260 | 554.2 |
| GILL | 0.418 | 0.227 | 274.5 |
| ThinkDiff-LVLM | **0.744** | **0.590** | **65.8** |

*Table 5.* Text-conditioned generation quality of ThinkDiff-LVLM on GenEval and DPG-Bench.

|  | Emu | SEED-LLaMA | Flux (Upperbound) | Ours |
|---|---|---|---|---|
| GenEval↑ | 3.25 | 35.4 | 65.1 | 39.1 |
| DPG-Bench↑ | 12.4 | 47.3 | 82.6 | 54.8 |

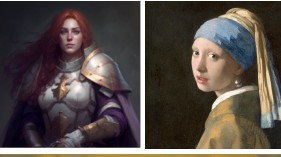
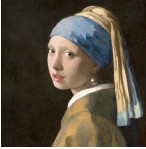
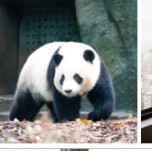
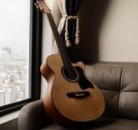
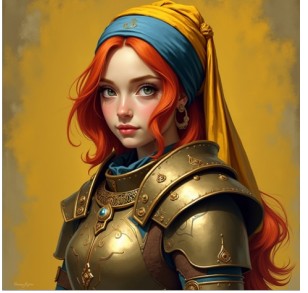
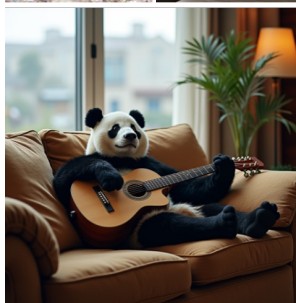

*Figure 8.* Results of **ThinkDiff-CLIP** composing two images. It creatively merge semantic details of both images. See more results in Appendix Figure 12.

of ThinkDiff-LVLM on GenEval (Ghosh et al., 2023) and DPG-Bench (Hu et al., 2024) in Table 5. Our method clearly outperforms Emu and SEED-LLaMA. Flux is used as our diffusion decoder. Therefore, its evaluation is the upperbound of our model. Please note that Geneval and DPG Bench focus on detailed text-prompt reconstruction, which does not fully evaluate ThinkDiff's strengths on multimodal reasoning and multimodal-to-image generation.

### 4.3. Evaluation Results of ThinkDiff-CLIP

We evaluate ThinkDiff-CLIP on various test cases to demonstrate its ability to semantically understand images and enable coherent composing of image and text modalities.

**Single image + text prompt.** Figure 6 and Appendix Figure 11 show results with a single image as input. FLUX Ultra (Forest, 2024b), possibly finetuned by reconstruction-based diffusion training, performs well in "copy-pasting" the input image (FLUX Ultra + I), but struggles to maintain coherence when an additional **text** prompt is included (FLUX Ultra + I + T). In contrast, ThinkDiff-CLIP excels at understanding the semantic details of the input image and effectively integrates both image and text to generate logically coherent outputs (Ours + I and Ours + I + T).

**Multiple images + text prompt.** ThinkDiff-CLIP is flexible and can handle multiple images and text prompts. As shown in Figure 8 and Appendix Figure 13, it can combine semantic details from two images in a reasonable and coherent manner. Figure 13 further demonstrates that with an additional text prompt (Ours + 2I + T), ThinkDiff-CLIP effectively incorporates the prompt into the generation.

These multimodal generation results highlight the advantage

of our vision-language training, which aligns multimodal features into a shared space, enabling flexible handling of complex multimodal understanding and composing tasks.

**Video generation.** ThinkDiff-CLIP is agnostic to diffusion decoders, making it versatile for integration with models like CogVideoX (Yang et al., 2024b), a text-to-video diffusion model. As shown in Appendix Figure 14, a background image is fed to the vision encoder and aligner network, along with a text prompt, to CogVideoX decoder. The model then generates a coherent video by seamlessly integrating images and text. This demonstrates ThinkDiff-CLIP's flexibility and broad applicability for multimodal generation tasks.

### 4.4. Ablation Study

**RMSNorm in the aligner network.** As discussed in Section 3.2, the RMSNorm layer and its initialization are critical for training convergence. Figure 7 compares training losses of three setups: without a RMSNorm layer, with default initialization, and with our final design. Without a RMSNorm layer or using default initialization, the training loss fails to converge while with our design, the loss converges to a reasonable value, leading to strong evaluation performance. This comparison validates the effectiveness of our design.

**Random masked training strategy.** As discussed in Section 3.3, we introduce a masked training strategy to address the "shortcut mapping" problem in ThinkDiff-LVLM training. In Table 3, we compare the 2-shot accuracy on CoBSAT benchmark for models trained with and without this strategy.

*Table 6.* Ablations of contrastive learning, data scale, different LVLMs, Janus Pro, and Flux Redux. ThinkDiff-LVLM uses Qwen2-VL-7b and 1.7M samples for training.

| | Color-I | Background-I | Style-I | Action-I | Texture-I | Color-II | Background-II | Style-II | Action-II | Texture-II |
|---|---|---|---|---|---|---|---|---|---|---|
| Contrastive | 0.414 | 0.244 | 0.140 | 0.202 | 0.230 | 0.347 | 0.346 | 0.235 | 0.258 | 0.231 |
| Janus pro | 0.403 | 0.234 | 0.378 | 0.462 | 0.338 | 0.313 | 0.319 | 0.283 | 0.549 | 0.264 |
| Flux Redux | 0.042 | 0.052 | 0.124 | 0.106 | 0.002 | 0.039 | 0.046 | 0.050 | 0.082 | 0.004 |
| 3.4M data | 0.632 | 0.374 | 0.233 | 0.484 | 0.323 | 0.469 | 0.573 | 0.354 | 0.523 | 0.281 |
| Internvl-2.5-8b | 0.326 | 0.108 | 0.104 | 0.261 | 0.111 | 0.278 | 0.308 | 0.163 | 0.495 | 0.137 |
| Qwen2-VL-72b | 0.656 | 0.363 | 0.359 | 0.361 | 0.375 | 0.458 | 0.617 | 0.411 | 0.538 | 0.338 |
| ThinkDiff-LVLM | 0.622 | 0.349 | 0.237 | 0.459 | 0.290 | 0.511 | 0.534 | 0.340 | 0.534 | 0.292 |

*Table 7.* Training resources and 4-shot accuracy. ThinkDiff-LVLM drastically reduces GPU usage and training time and improves accuracy from 0.192, 0.07, and 0.058 to 0.463.

| | GPU No. | Time / h | Average Acc. |
|---|---|---|---|
| SEED-LLaMA | 64 A100 | 216 | 0.192 |
| Emu | 128 A100 | 48 | 0.070 |
| GILL | 2 A6000 | 48 | 0.058 |
| ThinkDiff-LVLM | 4 A100 | 5 | **0.463** |

Without the random masked training, ThinkDiff-LVLM converges quickly but achieves inferior evaluation accuracy, indicating incomplete feature space alignment. In contrast, with the random masked training, the model achieves SoTA accuracy on the evaluation tasks. This validates the critical role of the random masked training for proper feature alignment in ThinkDiff-LVLM.

**Using generated tokens of LVLM.** As discussed in Section 3.3, ThinkDiff-LVLM uses deep features of generated tokens from the LVLM to effectively transfer reasoning information to diffusion decoders. In this study, we train a model using the deep features of input tokens of LVLM for alignment, with these features extracted from the final normalization layer of the LVLM. As shown in Table 3, using input token features for alignment leads to a significant performance drop, underscoring the critical role of generated tokens in successfully transferring reasoning capabilities.

**Training time and GPU usage.** Table 7 summarizes the training time, GPU requirements, and 4-shot average accuracy on CoBSAT for different methods. Our method drastically reduces GPU usage from 128 A100 GPUs to just 4 and cuts training time from 216 hours to only 5 hours. Meanwhile, it achieves a significant improvement in average accuracy, increasing from 0.192, 0.070, and 0.058 to an impressive 0.463. These results highlight the efficiency and effectiveness of our novel alignment paradigm.

**Comparison to contrastive learning.** We conduct an experiment with ImageBind-style (Girdhar et al., 2023) contrastive alignment. The input of LVLM is an image and a text prompt. It generates token features and text. The input of the T5 decoder is the LVLM's generated text. Instead

of using only one token similar to the original Imagebind, we extract 32 semantic tokens from both LVLM and T5 to compute the alignment loss. As shown in Table 6, our ThinkDiff achieves significant improvements on the CoBSAT benchmark over the ImageBind-style alignment.

**Different VLMs.** We use Qwen2-VL-7B as the LVLM by default, which supports interleaved image and text inputs. We ablates ThinkDiff-LVLM's performance with different LVLMs. As shown in Table 6, InternVL2.5-8B achieves a worse performance compared to Qwen2-VL-7B, indicating that a stronger LVLM can improve the alignment and accuracy. Moreover, with a more powerful LVLM Qwen2-VL-72B, ThinkDiff achieves a new SoTA on most tasks.

**Data scale.** We double the sample size to 3.4M to include more diverse datasets. We train a new model for the same steps. Table 6 shows that with more data, our model can generally improve the results in most tasks.

**Janus Pro and Flux Redux.** To evaluate Janus Pro (Chen et al., 2025) on CoBSAT, we implemented a two-step workaround since it lacks multimodal-to-image generation capabilities. Janus Pro converted multimodal inputs into intermediate textual descriptions, which were then processed through its text-to-image pipeline. As shown in Table 6, ThinkDiff outperforms Janus Pro in most tasks due to its alignment of powerful LVLMs and diffusion decoders, enabling superior multimodal reasoning and generation. We also evaluated the open-source Flux Redux (Labs, 2024), which supports image inputs on CoBSAT by organizing test cases into a single input image. As shown in Table 6, unlike ThinkDiff, Flux Redux performs poorly on CoBSAT, confirming its lack of reasoning capabilities.

## 5. Conclusion

We introduced ThinkDiff, a novel alignment paradigm equipping diffusion models with multimodal in-context reasoning of VLMs by vision-language training. ThinkDiff sets a new SoTA on the CoBSAT benchmark and excels in various reasoning tasks. Future work will address its limitations (Appendix A), and extend its capabilities to modalities like audio and video to develop any-to-any foundation models.

## Acknowledgements

This research is supported in part by HKUST-SAIL joint research funding, the Early Career Scheme of the Research Grants Council (RGC) of the Hong Kong SAR under grant No. 26202321 and HKUST Startup Fund No. R9253.

## Impact Statement

This paper proposed ThinkDiff, a novel alignment method that enhances text-to-image diffusion models by integrating multimodal in-context reasoning capabilities from vision-language models. By simplifying the alignment process between the VLM and diffusion decoder, ThinkDiff democratizes complex multimodal reasoning generation tasks and make them more accessible and efficient to train. ThinkDiff has potential applications across different fields, such as education, design, and creative industries. However, similar to other text-to-image diffusion models and large vision-language models, ThinkDiff could be potentially misused for generating misleading and harmful content. To mitigate these problems, it is essential to deploy the model responsibly and implement robust safeguards to prevent misuse.

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

# APPENDIX

## A. Limitation

Despite ThinkDiff's strong performance in reasoning generation tasks, several limitations remain for future work. First, while it substantially outperforms existing methods, ThinkDiff still encounters difficulties with certain complex cases. Enhancing reasoning accuracy may require stronger VLMs, better data quality, advanced diffusion models, and improved training strategies. Second, although this work primarily focuses on logical reasoning rather than preserving image fidelity, improving fidelity could expand its applications in tasks like image editing. Finally, more diverse evaluation tasks are needed to better assess reasoning performance and advance research in this area.

## B. Dataset Details

For ThinkDiff-LVLM, the training process requires images and their corresponding VLM-generated tokens. We randomly sample 1.7 million images from the CC3M (Sharma et al., 2018), CC12M (Changpinyo et al., 2021), and SBU (Ordonez et al., 2011) datasets. These images are preprocessed using Qwen2-VL, which generates detailed descriptions based on randomly selected text prompts from a predefined set. The generated text tokens and token features are stored for training the alignment. We generate 64 tokens for each data sample. Data processing is accelerated using the vLLM framework (Kwon et al., 2023).

For ThinkDiff-CLIP, the training utilizes images and their corresponding captions, sampled from a combination of CC3M (Sharma et al., 2018), CC12M (Changpinyo et al., 2021), SBU (Ordonez et al., 2011).

The predefined prompts for ThinkDiff-LVLM are designed to encourage the VLM to generate detailed descriptions of the image. Below is a list of the prompts we use, some of which are adapted from LLaVA (Liu et al., 2023).

- Describe the image concisely.

- Provide a brief description of the given image.

- Offer a succinct explanation of the picture presented.

- Summarize the visual content of the image.

- Give a short and clear explanation of the subsequent image.

- Share a concise interpretation of the image provided.

- Present a compact description of the photo's key features.

- Relay a brief, clear account of the picture shown.

- Render a clear and concise summary of the photo.

- Write a terse but informative summary of the picture.

- Create a compact narrative representing the image presented.

- Generate a prompt that can recreate the image in a 2D diffusion model.

- Provide a descriptive prompt to reproduce the given image using a diffusion model.

- Create a prompt suitable for a 2D diffusion model to generate the same image.

- Summarize the visual details as a prompt for a 2D diffusion model.

- Write a clear prompt to guide a 2D diffusion model in recreating the image.

**Evaluation on CoBSAT.** As described in Section 4.2, when evaluating ThinkDiff-LVLM on the CoBSAT dataset, we use an instruction prompt to guide Qwen2-VL to generate the next image based on multimodal inputs. Qwen2-VL is a vision-language model primarily designed to answer questions by text. It does not automatically know that we want it to generate the next image and we also do not finetune it for this specific task. Therefore, the instruction prompt is necessary. The instruction prompt used in our evaluation is:

- I give you several words and pictures. First, please analyse what the next picture is. Then give me a detailed diffusion prompt to describe the next picture. Please only provide me the detailed prompt and start the answer with 'Create an image'.

## C. More High-quality Results

### C.1. ThinkDiff-LVLM

Figure 9 and 10 demonstrate more high-quality results of ThinkDiff-LVLM on 2-shot evaluation in CoBSAT benchmark. ThinkDiff-LVLM can not only generate images with logically correct objects and attributes based on advanced reasoning, but also generate much higher-quality images than SEED-LLaMA (Ge et al., 2024), Emu (Sun et al., 2023), and GILL (Koh et al., 2024). These compared methods typically generate wrong images of lower quality.

### C.2. ThinkDiff-CLIP

Figure 11 shows more results with a single image (I) or a single image with a text prompt (I + T) as input. FLUX Ultra (Forest, 2024b) struggles to maintain coherence when an additional text prompt is included (FLUX Ultra + I + T) while ThinkDiff-CLIP excels at integrating both image and text to generate logically coherent images (Ours + I and Ours + I + T).

Figure 12 and 13 shows more results of our ThinkDiff-CLIP handling multiple images and text prompts. ThinkDiff-CLIP effectively combines semantic details from two input images in a coherent manner and seamlessly integrates text prompts to guide the generation, showcasing its flexibility and capability for complex multimodal tasks.

## D. Video Results of ThinkDiff-CLIP

As discussed in Section 4.3, ThinkDiff-CLIP can integrate CogVideoX (Yang et al., 2024b) model for text-to-video generation. Figure 14 demonstrates frames of video generation results, validating ThinkDiff-CLIP's flexibility and broad applicability for multimodal generation tasks.

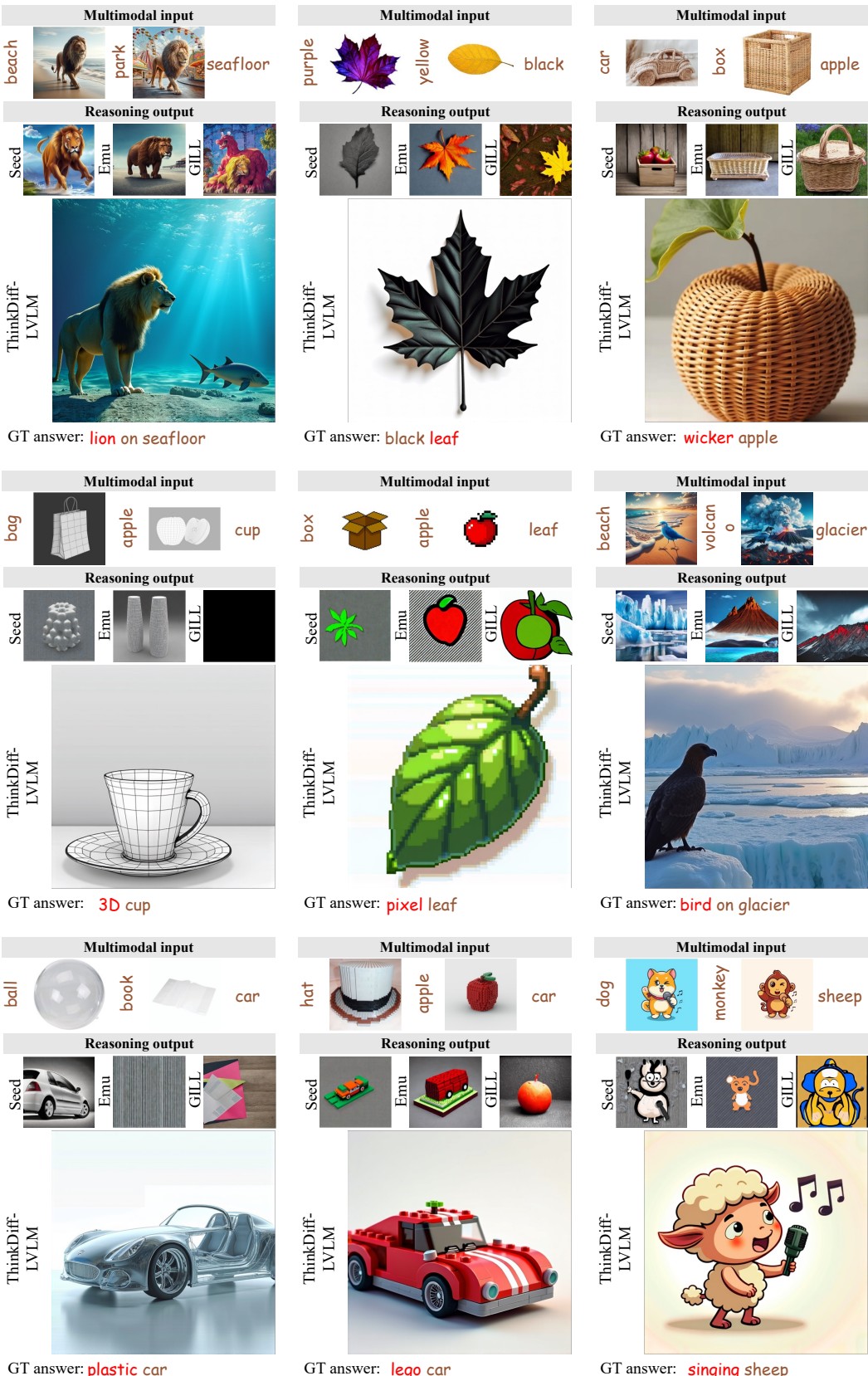

*Figure 9.* More 2-shot reasoning results of ThinkDiff-LVLM on CoBSAT benchmark.

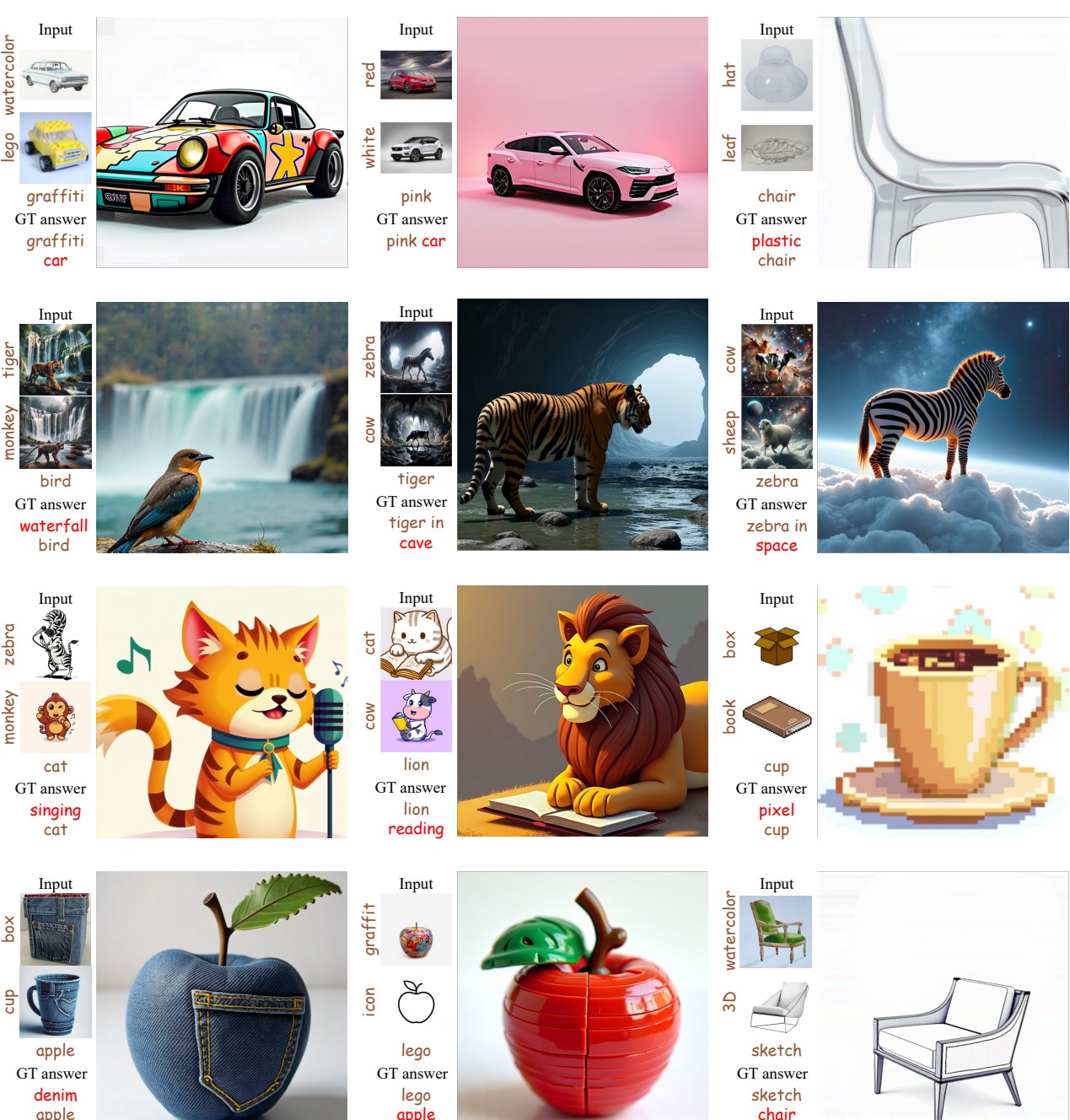

*Figure 10.* More 2-shot reasoning results of ThinkDiff-LVLM on CoBSAT benchmark.

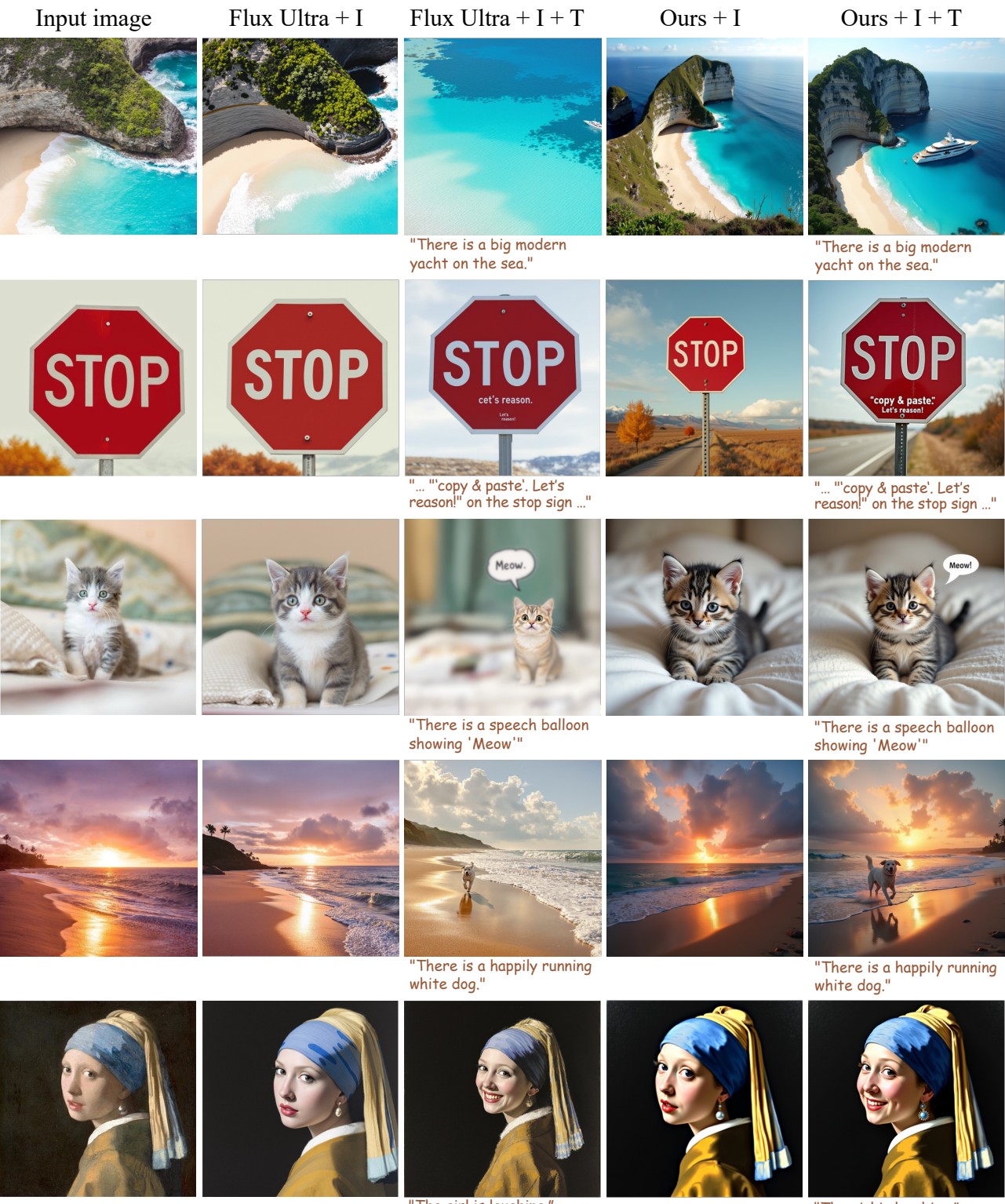

*Figure 11.* Generation results of a single image and a text prompt of ThinkDiff-CLIP.

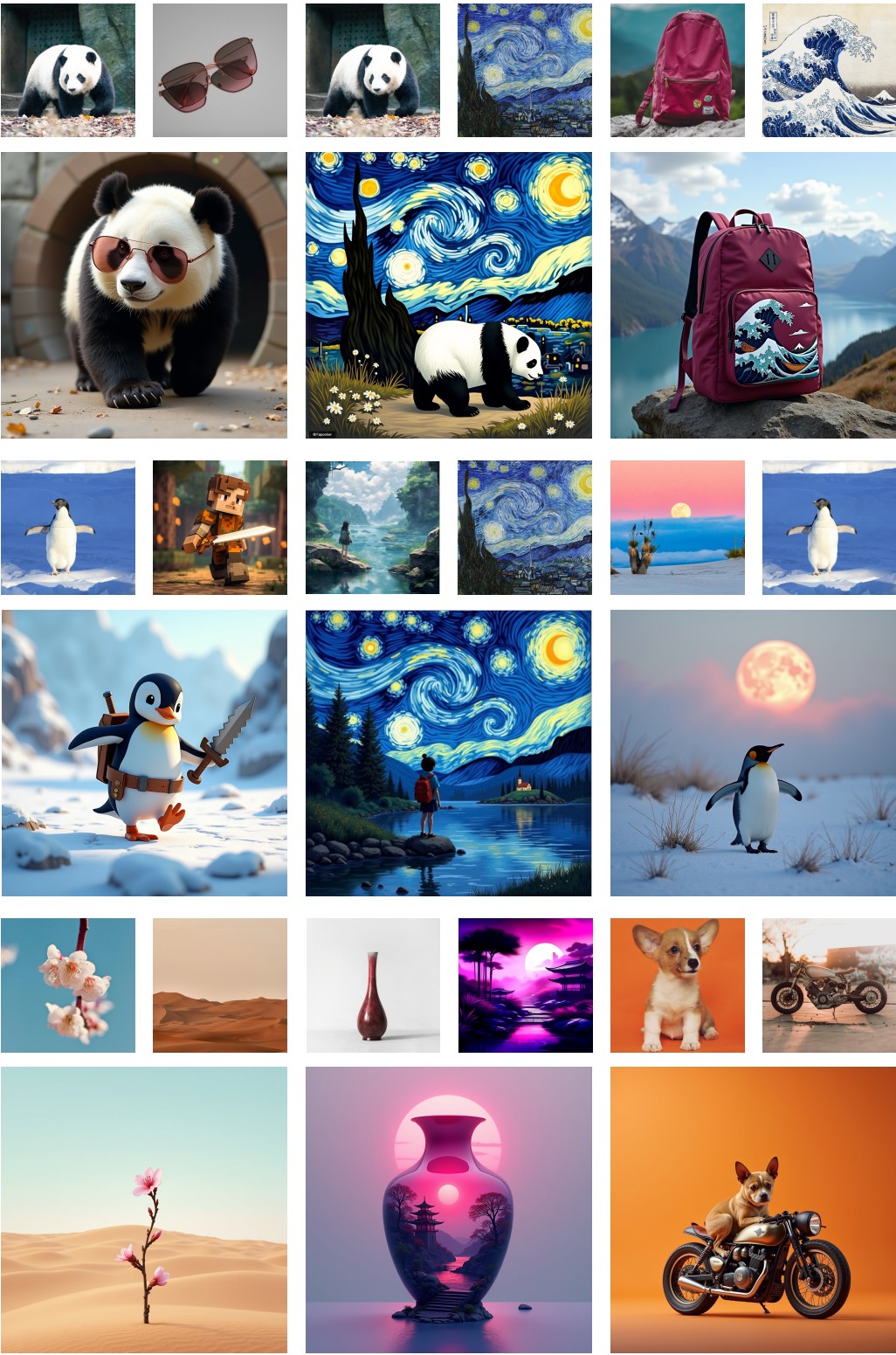

*Figure 12.* Multiple input image generation results of ThinkDiff-CLIP.

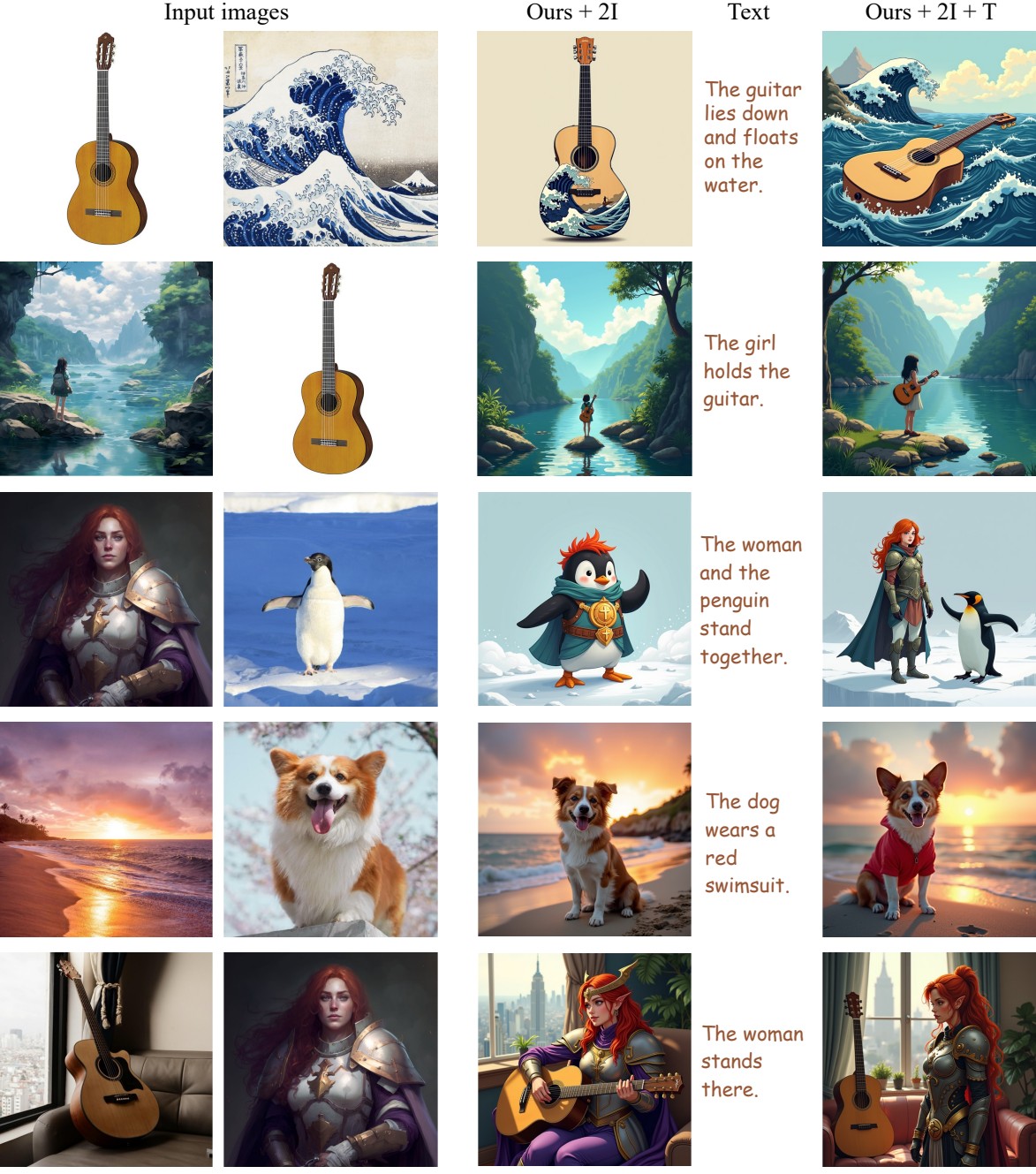

*Figure 13.* Generation results for multiple images (2I) and multiple images with a text prompt (2I + T) of ThinkDiff-CLIP.

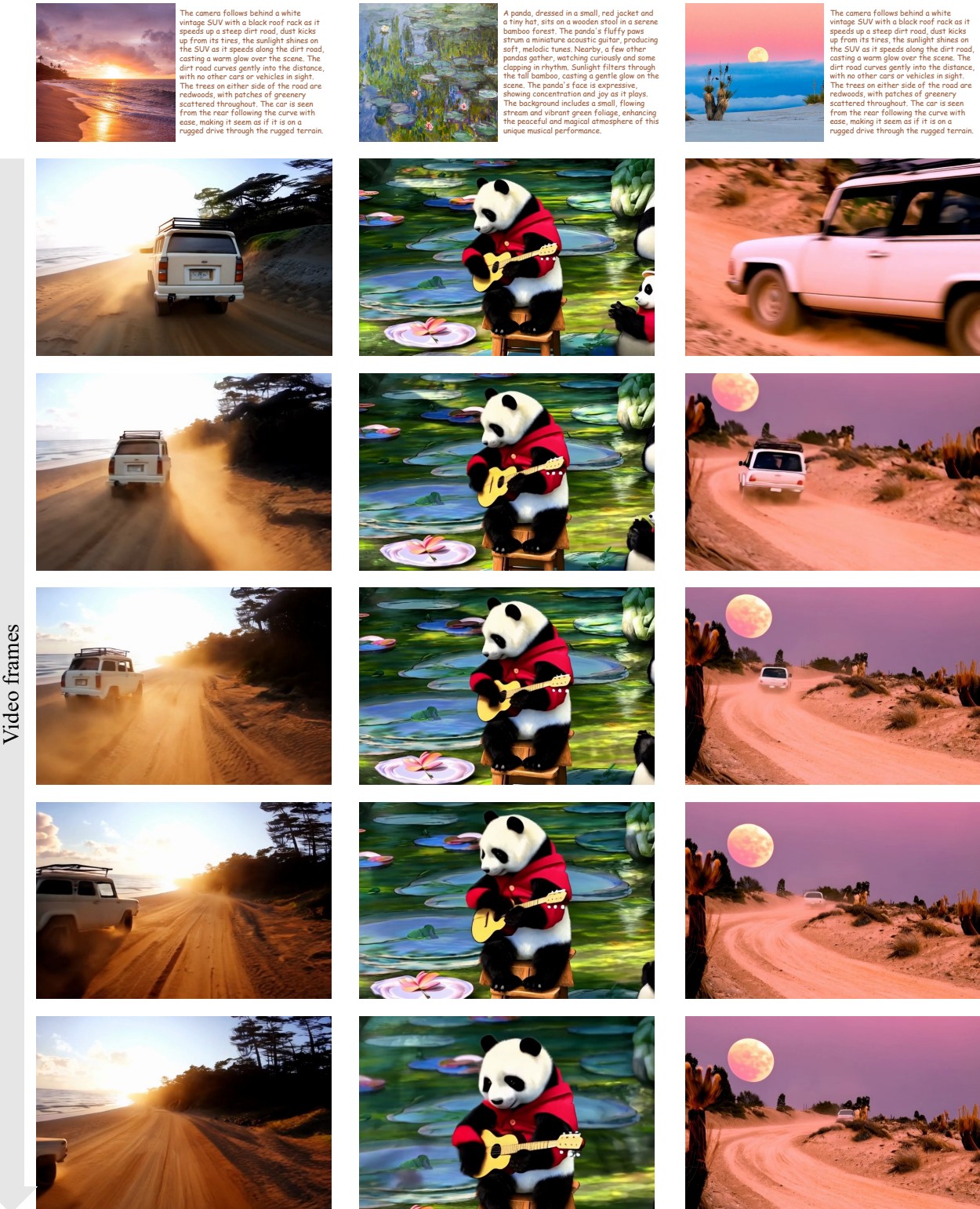

*Figure 14.* Image + text to video generation results of ThinkDiff-CLIP.

