# OpenReview forum: "I Think, Therefore I Diffuse: Enabling Multimodal In-Context Reasoning in Diffusion Models"
_ICML.cc/2025/Conference — ICML 2025 poster_

### Official Review · Reviewer_j31i · 2025-03-10

**Overall Recommendation:** 3

**Summary:**

This paper presents ThinkDiff, a framework that can efficiently and effectively align VLMs with diffusion models. Specifically, the framework design is inspired by the findings that latest diffusion models either use LLMs (e.g., T5) or CLIP as text encoders to guide the output image/video generation, so the paper aligns the embeddings from VLM with the encoded text embeddings with diffusion text encoder. Experiment results on in-context reasoning dataset (CoBSAT) has shown the effectiveness of the proposed ThinkDiff-VLM and ThinkDiff-CLIP models.

**Claims And Evidence:**

Yes, the claims that ThinkDiff is able to do in-context reasoning is well-supported.

**Essential References Not Discussed:**

The references looks complete to me.

**Experimental Designs Or Analyses:**

- The experiment for in-context reasoning ability on CoBSAT is sound. However, an important 2-step baseline that first uses the VLM to do in-context learning and output the answer in text format, then give this text as input to the diffusion text encoder is not included. This baseline is essential to prove that aligning VLM with diffusion model is better than first use VLM to do in-context reasoning then use diffusion to do generation. I would doubt the necessity of doing such VLM-Diffusion alignment as ThinkDiff if no justification on this point is provided.

- It would make the paper stronger if more evaluation benchmarks on image generation are included, such as GenEval, DPG-Bench.

**Methods And Evaluation Criteria:**

Yes, the paper evaluate the in-context reasoning ability on CoBSAT dataset.

**Other Comments Or Suggestions:**

NA

**Other Strengths And Weaknesses:**

- Strength: the idea of aligning VLM with the text embeddings of diffusion text encoder is novel

- Weakness: As discussed above, a major weakness of this paper is the lack of proof that why we need such alignment instead of just use VLM to do reasoning and use diffusion to do generation separately. If the authors cannot provide a convincing benefits, the necessity of such alignment could be doubtful.

**Questions For Authors:**

NA

**Relation To Broader Scientific Literature:**

Prior works on diffusion models usually focus on generating high-fidelity images/videos, while not focusing too much on the reasoning ability. There are some recent concurrent works, such as LanDiff [1], start to explore the same direction of adding reasoning abilities to diffusion models.


[1] Yin, Aoxiong, et al. "The Best of Both Worlds: Integrating Language Models and Diffusion Models for Video Generation." arXiv preprint arXiv:2503.04606 (2025).

**Theoretical Claims:**

NA

---

> ### Author Rebuttal · Authors · 2025-04-01
>
> **Reviewer j31i**
>
> We thank Reviewer j31i for the insightful comments and suggestions. We provide our responses as follows and will add additional literature in the reference.
>
> >**Q1: Why alignment?**
>
> There are several advantages in terms of the alignment compared to text-based methods on the multimodal generation task. We elaborate on them below:
>
> 1. An important aspect of multimodal generation is to capture multimodal information in the generator. The aligner in our method is differentiable. It can convey detailed multimodal information that cannot be expressed by texts effectively. However, the text-based methods discard the detailed multimodal information in the middle.
> 2. The model is an end-to-end model. The submodels in it can be optimized together to benefit each other.
> 3. The features passed through the aligner are more representative and can convey more dense spatial context information than sparse texts.
>
> > **Q2: Text-based evaluation.**
>
> The CoBSAT benchmark mainly focuses on the correctness of logical reasoning. To better validate the benefits of the alignment that can convey more multimodal information, we evaluate the image-condition generation of ThinkDiff-LVLM on COCO, as shown in the table below. The input of ThinkDiff-LVLM is an image. The input of the text-based method is the texts generated by the LVLM. As shown in this table, the alignment performs better on the CLIP image metric (CLIP-I) and the FID score. This experiment further validates that with the proposed differentiable aligner and dense representations, more multimodal information can be passed to the diffusion decoder, which is important for multimodal generation tasks.
>
> | Metric      | CLIP-I$\uparrow$ | FID$\downarrow$   |
> |-------------|------------|-----------|
> | Alignment   | **0.744**  | **65.8**  |
> | Text-based  | 0.728      | 66.3      |
>
>
>
>
>
> > **Q3: Evaluation benchmarks.**
>
> We evaluate on more datasets such as the COCO, GenEval and DPG-Bench.
>
> The GenEval:
>
> | Model       | Emu   | SEED-LLaMA | Ours      |
> |-------------|-------|------------|-----------|
> | GenEval$\uparrow$ | 3.25  | 35.35      | **39.13** |
>
> The DPG-Bench:
>
> | Model       | Emu   | SEED-LLaMA | Ours      |
> |-------------|-------|------------|-----------|
> | DPG-Bench$\uparrow$ | 12.4  | 47.3       | **54.8**  |
>
> The COCO results are in Q2 and Reviewer BZ6o Q1.
>
> Compared to the previous arts, ThinkDiff-LVLM achieves notable gains in all benchmarks.

---

> > ### Comment · Reviewer_j31i · 2025-04-07
> >
> > Thanks for the authors' response! One last question about the response in Q3. It seems that the scores on GenEval and DPG-Bench are still much lower than other recent unified models (januspro, show-o, etc). As a reference, Fig1(b) in JanusPro shows the GenEval and DPG-Bench scores on SDXL, PixArt-Alpha, and SDv1.5. To at least show that the proposed alignment works better than text-based evaluation on the underlying diffusion model you used (which is FLUX), could the author provide a evaluation result with FLUX on these two benchmark datasets (similar as Fig1(b) in JanusPro)?
> >
> > Chen, Xiaokang, et al. "Janus-pro: Unified multimodal understanding and generation with data and model scaling." arXiv preprint arXiv:2501.17811 (2025).

---

> > > ### Author Response · Authors · 2025-04-09
> > >
> > > ## Reviewer j31i
> > >
> > > We sincerely thank Reviewer j31i for the insightful feedback and follow-up questions, and provide detailed responses to the new comments.
> > >
> > > >**Q4: General Remark**
> > >
> > > Our proposed ThinkDiff enables **multimodal reasoning ability** for pretrained image generation models through efficient alignment. Below, we clarify its advantages over FLUX and Janus Pro.
> > >
> > > **Compared to FLUX:**
> > >
> > > FLUX focuses on reconstructing text into images, whereas ThinkDiff performs multimodal reasoning generation.
> > >
> > > **Compared to Janus Pro:**
> > >
> > > (1) ThinkDiff is a general framework that enhances various pretrained generation models (e.g., text-to-image, text-to-video) with reasoning capabilities.
> > >
> > > (2) ThinkDiff achieves superior multimodal reasoning with fewer computational resources. Janus Pro requires 256 A100 GPUs for 14 days while ThinkDiff requires just 5 hours on 4 A100 GPUs.
> > >
> > > (3) ThinkDiff supports multimodal-to-image generation based on in-context reasoning. As shown in Janus Pro's paper, code, and GitHub issues (#144 in deepseek-ai/Janus), Janus Pro is limited to text-to-image generation and does not support multimodal image generation.
> > >
> > > > **Q5: Geneval and DPG Bench**
> > >
> > > While **Geneval and DPG Bench** are valuable for evaluating text-to-image diffusion models, they are not designed for **multimodal reasoning generation**, the core strength of ThinkDiff. They evaluate text-prompt fidelity but lack support for multimodal inputs or reasoning.
> > >
> > > In contrast, the **CoBSAT** is explicitly designed to assess **multimodal reasoning generation**. It can highlight ThinkDiff's novel contributions.
> > >
> > > >**Q6: Janus Pro on reasoning benchmark.**
> > >
> > > To evaluate Janus Pro on CoBSAT, we implemented a two-step workaround since it lacks multimodal-to-image generation capabilities. Janus Pro converted multimodal inputs into intermediate textual descriptions, which were then processed through its text-to-image pipeline. The results are summarized below:
> > >
> > > **Janus Pro on CoBSAT:**
> > >
> > > |       | Color-I   | Background-I | Style-I   | Action-I  | Texture-I | Color-II  | Background-II | Style-II  | Action-II | Texture-II |
> > > |-|-|-|-|-|-|-|-|-|-|-|
> > > | Janus | 0.403     | 0.234        | **0.378** | **0.462** | **0.338** | 0.313     | 0.319         | 0.283     | 0.549     | 0.264      |
> > > | Ours  | **0.638** | **0.362**    | 0.254     | 0.434     | 0.317     | **0.610**  | **0.590**      | **0.432** | **0.664** | **0.332**  |
> > >
> > > **Key Observations:**
> > >
> > > (1) ThinkDiff outperforms Janus Pro in most tasks due to its alignment of powerful LVLMs and diffusion decoders, enabling superior multimodal reasoning and generation.
> > >
> > > (2) Janus Pro struggles to balance reasoning and high-quality image generation and lacks native multimodal-to-image capabilities.
> > >
> > > (3) While Janus Pro excels on text-centric benchmarks (Geneval, DPG Bench), these benchmarks do not evaluate **multimodal reasoning** or **multimodal-to-image generation**, which are ThinkDiff's key strengths.
> > >
> > >
> > > >**Q7: Evaluation of Flux and Flux Redux**
> > >
> > > We evaluated Flux on GenEval and DPG-Bench, including Janus Pro's results for reference:
> > >
> > > **GenEval Results:**
> > >
> > > | Emu   | SEED-LLaMA | Janus Pro | Flux (Upperbound) | Ours |
> > > |-|-|-|-|-|
> > > | 3.25  | 35.35      | 80.0        | 65.13            | 39.13 |
> > >
> > > The results of Flux can be cross-validated by Table 2 in the literature: 1.58-bit FLUX, Chenglin Yang, et. al.
> > >
> > > **DPG-Bench Results:**
> > >
> > > | Emu   | SEED-LLaMA | Janus Pro | Flux (Upperbound) | Ours |
> > > |-|-|-|-|-|
> > > | 12.4  | 47.3       | 84.2      | 82.6             | 54.8 |
> > >
> > >
> > > We also evaluated the open-source Flux Redux which supports image inputs on CoBSAT by organizing test cases into a single input image. Results are below:
> > >
> > > **Flux Redux on CoBSAT:**
> > >
> > > |       | Color-I | Background-I | Style-I | Action-I | Texture-I | Color-II | Background-II | Style-II | Action-II | Texture-II |
> > > |-|-|-|-|-|-|-|-|-|-|-|
> > > | Flux Redux | 0.042   | 0.052        | 0.124   | 0.106   | 0.002     | 0.039    | 0.046         | 0.050     | 0.082    | 0.004      |
> > > | Ours  | **0.638** | **0.362**    | **0.254** | **0.434** | **0.317** | **0.61** | **0.59**      | **0.432** | **0.664** | **0.332**  |
> > >
> > >
> > > We can observe from these three tables that:
> > >
> > > (1) Flux performs well on text-to-image tasks, consistent with its reconstruction-based design.
> > >
> > > (2) FLUX sets the upper bound for ThinkDiff on Geneval and DPG Bench, as ThinkDiff builds on FLUX.
> > >
> > > (3) Unlike ThinkDiff, Flux Redux performs poorly on CoBSAT, confirming its lack of reasoning capabilities. Geneval and DPG Bench focus on detailed text-prompt reconstruction, which does not fully evaluate ThinkDiff's strengths.
> > >
> > > (4) ThinkDiff focuses on multimodal reasoning generation. For reconstruction-based tasks, ThinkDiff may slightly alter input text prompts to enhance the semantic richness, which can impact performance on text-centric benchmarks. However, the core strength of ThinkDiff lies in multimodal reasoning generation. For reconstruction-based text-to-image tasks, the original FLUX can still be used.

---

### Official Review · Reviewer_fnpq · 2025-03-10

**Overall Recommendation:** 4

**Summary:**

This paper introduces **ThinkDiff**, a novel alignment paradigm that enhances text-to-image diffusion models with **multimodal in-context reasoning** capabilities. Instead of traditional pixel-level reconstruction-based fine-tuning, the authors propose aligning vision-language models (VLMs) with the decoder of a **large language model (LLM)**, leveraging vision-language training as a **proxy task**. The key insight is that **LLM decoders and diffusion model decoders share the same input feature space**, allowing the transfer of reasoning capabilities without requiring complex multimodal reasoning datasets.

**Claims And Evidence:**

The claims made in the paper are generally well-supported by empirical evidence. The authors provide:

- **Quantitative results** demonstrating large accuracy improvements on the CoBSAT benchmark compared to prior methods.
- **Ablation studies** validating key components of ThinkDiff, such as the importance of aligning generated token features instead of input token features.
- **Comparisons to existing approaches**, showing that ThinkDiff not only outperforms but also requires significantly fewer resources.

However, while the results are strong, further validation on **diverse datasets and real-world applications** would reinforce the claims regarding generalizability.

**Essential References Not Discussed:**

The paper thoroughly cites relevant prior work but could benefit from discussing:

- **Work on vision-language fine-tuning paradigms** (e.g., InstructBLIP, Kosmos-G).
- **More literature on feature space alignment in deep learning** (e.g., feature alignment in contrastive learning).

Explicitly comparing ThinkDiff’s **alignment approach** to **existing multimodal fusion techniques** could further strengthen the paper.

**Experimental Designs Or Analyses:**

The experimental design is well-structured, with:

- **Comparisons against multiple baselines**, including SEED-LLaMA, Emu, and GILL.
- **Ablation studies** examining the effectiveness of masked training, LVLM-generated tokens, and RMSNorm initialization.
- **Efficiency comparisons**, showing that ThinkDiff achieves better performance with significantly lower computational costs.

However, the dataset used for training is relatively **small (~1.7M images from CC3M, CC12M, and SBU)**. Additional results on **larger and more diverse datasets** could provide stronger evidence of generalizability.

**Methods And Evaluation Criteria:**

The proposed methods and evaluation criteria are appropriate for the problem:

- The **CoBSAT benchmark** is a well-suited dataset for evaluating multimodal reasoning in diffusion models.
- The evaluation method follows a structured **2-shot and 4-shot reasoning approach**, ensuring fair comparisons.
- **Ablation studies** confirm the contribution of the core components of ThinkDiff.

However, the work mainly focuses on **image generation tasks**. It would be valuable to evaluate the method on **other multimodal reasoning tasks**, such as **captioning or VQA**, to assess broader applicability.

**Other Comments Or Suggestions:**

- The **writing is clear and well-structured**, making it easy to follow the key ideas.
- The **figures effectively illustrate the concept of multimodal in-context reasoning**.
- Additional **comparison to multimodal alignment techniques** (e.g., how ThinkDiff differs from IP-Adapter) would be useful.

**Other Strengths And Weaknesses:**

#### **Strengths**:

1.  **Novel multimodal alignment framework**, enabling in-context reasoning in diffusion models.
2.  **Significant improvement over baselines**, achieving SOTA performance on CoBSAT.
3.  **Computational efficiency**, requiring fewer GPUs and training hours than competing methods.
4.  **Clear experimental design**, including thorough ablation studies and efficiency comparisons.

#### **Weaknesses**:

1.  **Limited evaluation on real-world datasets**—performance on **other multimodal tasks** like captioning or VQA is unclear.
2.  **Theoretical analysis of feature space alignment is minimal**—a deeper mathematical justification could be beneficial.
3.  **Only tested on a single benchmark (CoBSAT)**—generalizability to other multimodal datasets remains an open question.

**Questions For Authors:**

1. **Generalization beyond CoBSAT**:
   - How well do you expect ThinkDiff to generalize to **other multimodal tasks** beyond image generation (e.g., estimation, segment)?
2. **Scalability to other models**:
   - How does ThinkDiff perform with **other diffusion models** (e.g., SDXL, DeepFloyd IF)?
   - Does scaling the LVLM improve results, or does performance saturate?
3. **Feature space alignment analysis**:
   - Have you conducted any **quantitative analysis** on the feature space alignment between the **VLM, LLM decoder, and diffusion decoder**?
   - Could a **contrastive learning-based loss** further improve alignment?
4. **Ablation on different VLMs**:
   - Did you test **other VLMs** beyond Qwen2-VL and CLIP?
   - Would a **stronger LVLM (e.g., GPT-4V)** further improve results?
5. **Failure cases and limitations**:
   - Can you provide qualitative examples of **failure cases** where ThinkDiff struggles?
   - What are the **main sources of error**, and how could future work address them?

**Relation To Broader Scientific Literature:**

This work builds on several key areas:

- **Text-to-image diffusion models** (e.g., Stable Diffusion, Imagen).
- **Multimodal large language models (LLMs)** (e.g., Flamingo, SEED-LLaMA).
- **Vision-language models (VLMs)** (e.g., CLIP, BLIP-2).
- **Multimodal reasoning benchmarks** (e.g., CoBSAT).

ThinkDiff **extends prior work** by introducing a novel alignment paradigm that allows diffusion models to reason over multimodal inputs rather than simply reconstructing images.

**Theoretical Claims:**

The paper does not focus heavily on theoretical derivations but instead introduces a **novel alignment framework**. The core theoretical claim is that aligning **VLMs with LLM decoders naturally aligns them with diffusion model decoders** due to their shared input feature space.

This is a **reasonable assumption**, but additional theoretical validation or **formal analysis of feature space alignment** could further strengthen the claim.

---

> ### Author Rebuttal · Authors · 2025-04-01
>
> **Reviewer fnpq**
>
> We thank reviewer fnpq for the insightful comments and suggestions. We address major concerns as follows and will add additional literature in the reference.
>
>
> >**Q1: Ablation on different VLMs.**
>
> We use Qwen2-VL-7B which supports interleaved image and text inputs as the LVLM. The table below further ablates ThinkDiff-LVLM's performance with different LVLMs.
> InternVL2.5-8B achieves a worse performance compared to Qwen2-VL-7B, indicating that a stronger LVLM can improve the alignment and accuracy. Moreover, with a more powerful LVLM Qwen2-VL-72B, ThinkDiff achieves a new SoTA on most tasks.
>
> | Model            | Color-I | Background-I | Style-I | Action-I | Texture-I | Color-II | Background-II | Style-II | Action-II | Texture-II |
> |----|---|----|---|----|---|----|---|----|---|-----|
> | Internvl-2.5-8b  | 0.326   | 0.108        | 0.104   | 0.261    | 0.111     | 0.278    | 0.308         | 0.163    | 0.495     | 0.137      |
> | Qwen2-VL-7b      | 0.622   | 0.349        | 0.237   | **0.459**| 0.290      | **0.511**| 0.534         | 0.340     | 0.534     | 0.292      |
> | Qwen2-VL-72b     | **0.656**| **0.363**   | **0.359**| 0.361    | **0.375** | 0.458    | **0.617**     | **0.411**| **0.538** | **0.338**  |
>
>
> >**Q2: Data scale.**
>
> We double the sample number to 3.4M to include more diverse datasets. We train a new model for the same steps. The table below shows with more data, our model can generally improve the results in most tasks.
>
> | Data | Color-I | Background-I | Style-I | Action-I | Texture-I | Color-II | Background-II | Style-II | Action-II | Texture-II |
> |----|---|---|----|-----|---|---|---|---|----|----|
> | 1.7M  | 0.622   | 0.349        | **0.237** | 0.459    | 0.29      | **0.511**| 0.534         | 0.340     | **0.534** | **0.292**  |
> | 3.4M  | **0.632**| **0.374**   | 0.233   | **0.484**| **0.323** | 0.469    | **0.573**     | **0.354**| 0.523     | 0.281      |
>
>
> >**Q3: Contrastive loss.**
>
> In **Q1** and **Q2** of Reviewer 37KH, we use the contrastive loss similar to the ImageBind. It has some disadvantages compared to our method and is inferior in accuracy. However, it still shows potential to effectively combine the explicit contrastive alignment and our implicit alignment in future work.
>
>
> >**Q4: Real-world application.**
>
> Our method not only can handle the tasks defined by the CoBSAT benchmark but also can handle different general tasks. To demonstrate this point, we further evaluate our model on general generation capabilities on other benchmarks, i.e., COCO, GenEval, and DPG-bench. Our method shows clearly better results compared to other competitors. Please refer to the COCO table in Reviewer BZ6o **Q1**, the GenEval, and the DPG-bench tables in Reviewer j31i **Q3**.
>
> >**Q5: Other multimodal fusion techniques.**
>
> The Flux-pro actually uses a fusion method similar to IP-Adapter which injects image features by attention. As shown in Paper Figures 6, 11, and 13 in the main paper, our method clearly shows advantages in coherently composing different multimodal instructions.
>
> >**Q6. Formal analysis of feature space alignment.**
>
> The alignment quality of ThinkDiff is implicitly evaluated by the accuracy and consistency of the reasoning benchmark and composing benchmark. Since our alignment is implicit, it is important but not very straightforward to directly analyze the feature space. To investigate the theoretical derivations of the alignment, we may rely on the development of vision-language training in VLMs. We humbly leave this for future work.
>
> >**Q7: Other multimodal reasoning tasks.**
>
> In this paper we mainly target multimodal generation tasks. The LVLM in our model supports captioning or VQA tasks but the whole method itself does not target them. However, our alignment is a general alignment method and can be possibly applied to other multimodal tasks. This is a very interesting direction for future research.
>
> >**Q8: Other diffusion models.**
>
> Once aligned, our ThinkDiff can be applied to other models that use T5 as encoder in our experiments. For example, beyond FLUX, we also applied ThinkDiff to CogvideoX in Appendix Figure 14, where a coherent video is generated by seamlessly integrating images
> and text. This demonstrates ThinkDiff's flexibility. Extending ThinkDiff to even more diffusion models is straightforward and is left as future work due to the tight schedule.
>
>
> >**Q9: Failure cases.**
>
> This anonymous link (https://anonymous.4open.science/r/anonymous-4DF1/failure_case.png) shows two failure cases on CoBSAT. We think the main sources are from two aspects. One is the imperfect of LVLM which gives wrong reasoning, such as the "cow in the desert" case. Using a more powerful VLM can address this problem. The other is the imperfect alignment that does not accurately condition diffusion models, such as the second "white apple" case. We think a possible way to effectively address this problem is to include high-quality datasets for end-to-end training.

---

### Official Review · Reviewer_BZ6o · 2025-03-14

**Overall Recommendation:** 3

**Summary:**

This paper proposes "ThinkDiff", a novel method to incorporate VLMs in text-to-image generation pipelines with the goal to improve multimodal understanding and in-context reasoning capabilities. The key lies in aligning the VLM outputs with the diffusion decoder input space, which is done by using the corresponding LLM Decoder as a proxy signal for alignment. This results in a lightweight training algorithm and empirical results demonstrate a) significant improvements for in-context generation on CoBSAT and b) qualitative performance on compositional tasks.

**Claims And Evidence:**

Most of the claims in the paper are well-supported by their experiments. However, there is no quantitative evaluation for the quality of generated images as well as analysis of inference compute, which needs to be addressed (see Methods And Evaluation Criteria).

**Essential References Not Discussed:**

N/A

**Experimental Designs Or Analyses:**

The experimental design is sound but not comprehensive enough (see Methods And Evaluation Criteria).

**Methods And Evaluation Criteria:**

Certain experiments and benchmarks are missing from the current manuscript.

1. Despite strong results on multimodal in-context generation, the paper lacks in terms of quantitative evaluation outside of the in-context generation settings. For instance, quantitative evaluation of quality of image generation (ex. FID/CLIP scores on zero-shot generation on COCO or subject driven generation on DreamBench) as well as a comparison against contemporaries is missing. However, these are important to ascertain if there are any potential drawbacks arising from the proposed framework and its focus on reasoning.
2. While low compute requirements for training is desirable, inference cost is arguably a bigger decision factor for model adoption. Therefore, the paper needs to include an analysis for the inference costs (especially due to the additional multimodal processing in VLMs) to complement their training cost comparisons.
3.  The paper proposes two models - ThinkDiffLVLM and ThinkDiffCLIP. However, the first is only evaluated on in-context settings while the other only on compositional settings without any specific reasoning for the choice. Evaluation of both methods on both settings is needed to understand their respective strengths and weaknesses.

Nevertheless, the framework is simple and effective, and my overall opinion of the paper is generally positive. I believe that addressing these concerns will serve to strengthen the manuscript.

**Other Comments Or Suggestions:**

N/A

**Other Strengths And Weaknesses:**

The paper is well structured and easy to follow, while the motivation and approach seem sound.

**Questions For Authors:**

N/A

**Relation To Broader Scientific Literature:**

The method presents a novel approach to incorporate a VLM's multimodal reasoning ability for in-context image generation with the added advantage of a lightweight training pipeline. The resulting model achieves significant gains on the CoBSAT dataset, compared to existing approaches. With some additional empirical validation, it can be a good contribution for the research community.

**Theoretical Claims:**

N/A

---

> ### Author Rebuttal · Authors · 2025-04-01
>
> **Reviewer BZ6o**
>
> We thank Reviewer BZ6o for the insightful comments and suggestions. We provide our responses below.
>
> >**Q1: Generation quality.**
>
> We evaluate the general image-conditioned generation of ThinkDiff-LVLM on 1k images in COCO. The models are conditioned by an image in the experiment. We show the FID, the CLIP image metric (CLIP-I), and the CLIPScore (CLIP-T) in the table below. Our method can achieve much better performance than existing competitors such as SEED-LLaMA, Emu, and GILL.
>
>
> | Model            | CLIP-I$\uparrow$   | CLIP-T$\uparrow$     | FID$\downarrow$        |
> |------------------|------------|------------|------------|
> | SEED-LLaMA       | 0.695     | 0.546     | 71.7      |
> | Emu              | 0.443     | 0.260     | 554.2     |
> | GILL             | 0.418     | 0.227     | 274.5     |
> | ThinkDiff-LVLM   | **0.744** | **0.590** | **65.8**  |
>
> >**Q2: Inference time.**
>
> ThinkDiff-LVLM replaces the T5 encoder in Flux with an LVLM. The LVLM model is highly optimized in the community framework such as vLLM and adds minimal latency to the inference. Qwen2-VL-7B model takes less than 0.2 seconds for one prompt that consists of both images and texts. Although the T5 encoder (typically costs 0.05 seconds for one prompt) is faster, they both are a marginal overhead compared to the diffusion process of Flux (typically over 2.5 seconds).
>
> >**Q3: ThinkDiff-LVLM vs ThinkDiff-CLIP and their evaluation.**
>
> ThinkDiff-LVLM can handle different tasks including general image-conditioned generation, similar to ThinkDiff-CLIP. As shown in **Q1**, we evaluate ThinkDiff-LVLM on the image-conditioned generation task. Our method obtains better CLIP-I, CLIP-T, and FID scores compared to SEED-LLaMA, Emu, and GILL. This shows the general applicability of ThinkDiff-LVLM beyond the reasoning.
>
> On the contrary, ThinkDiff-CLIP is limited by the CLIP image encoder and the T5 encoder. Therefore, in our experiments in the main paper, we already observed that ThinkDiff-CLIP cannot handle very complex logical questions as ThinkDiff-LVLM does, but it has strong capabilities for multimodal composing.
>
> The above additional evaluation further shows their respective strengths and weaknesses.

---

### Official Review · Reviewer_37KH · 2025-03-18

**Overall Recommendation:** 3

**Summary:**

The paper enables diffusion models to perform in-context reasoning across images and text, rather than just reconstructing pixel information. The paper shows two variants --- LVLMs and CLIP. The images generated are of good quality and obtain state of the art performance on various measures.

**Claims And Evidence:**

Yes, the claims are supported by the evidence.

**Essential References Not Discussed:**

NA

**Experimental Designs Or Analyses:**

The experiments look correct to me.

**Methods And Evaluation Criteria:**

Yes, the proposed method correctly reflects the application at hand.

**Other Comments Or Suggestions:**

Please see questions for authors.

**Other Strengths And Weaknesses:**

Please see questions for authors.

**Questions For Authors:**

I list down the strengths and the weaknesses of the work here

Strength

- The paper proposes a method that can effectively handle reasoning in diffusion model.
- The experimental gain is significant, and good for the community to build on.

Weakness

- The major concern appears to be the key aspect of the model. Overall, I feel the method is leveraging the strong powers of the base model---LVLMs or CLIP and use that to learn a multimodal aligner. This idea has been used previously in video and text representation learning, and this work proposes to extend that to multimodal composition. It is not clear why the authors could not use any other way to learn the joint embedding space. For example, ImageBind, Rohit Girdhar et al. performs this step in the image space. What is the key insight that enables this method to do better beyond existing work?

- Same as above, ImageBind by Girdhar et al. is not compared against in the experiments. Why can it not be adapted to the authors' use case? The overall idea is for the model to be compared against a variety of inputs.

Overall, I am leaning towards acceptance, but the distinctions and experiments could be made more robust and convincing. .

**Relation To Broader Scientific Literature:**

The paper is related to the broader scientific literature.

**Theoretical Claims:**

There are no theoretical claims in the paper.

---

> ### Author Rebuttal · Authors · 2025-04-01
>
> **Reviewer 37KH**
>
> We thank Reviewer 37KH for the insightful comments and suggestions. We provide our responses below.
>
>
> >**Q1. Experiment with Imagebind.**
>
> We conduct an experiment with ImageBind-style alignment to align the LVLM decoder and the T5 encoder. The input of LVLM is an image and a text prompt. It generates token features and text. The input of the T5 decoder is the LVLM's generated text. Instead of using only one token similar to the original Imagebind, we extract 32 semantic tokens from both LVLM and T5 to compute the alignment loss. This gives stronger capacity to the alignment. As shown in the table below, our ThinkDiff achieves significant improvements on the CoBSAT benchmark over the ImageBind-style alignment.
>
> | Model      | Color-I | Background-I | Style-I | Action-I | Texture-I | Color-II | Background-II | Style-II | Action-II | Texture-II |
> |------------|:-------:|:------------:|:-------:|:--------:|:---------:|:--------:|:-------------:|:--------:|:---------:|:----------:|
> | **ImageBind**  | 0.414   | 0.244        | 0.140   | 0.202    | 0.230     | 0.347    | 0.346         | 0.235    | 0.258     | 0.231      |
> | **Ours**       | **0.622**  | **0.349**        | **0.237**   | **0.459**    | **0.290**     | **0.511**    | **0.534**         | **0.340**    | **0.534**     | **0.292**      |
>
>
> >**Q2. Advantages over ImageBind.**
>
> While the contrastive alignment in Imagebind shows promising results in multimodal alignment, ThinkDiff introduces more advantages:
>
>
> 1. ThinkDiff exploits the capabilities and knowledge of the LVLM decoder and the LLM (T5) decoder for the alignment, which enables data-efficient and training-efficient alignment, and has been validated in vision-language research areas. On the contrary, Imagebind needs web-scale datasets to train from scratch.
>
> 2. ThinkDiff provides a fine-grained and flexible alignment method. Since it does not compute element-wise distances, the tokens of different modalities can have different lengths. This also supports training strategies such as masked training. The alignment is not element-wise minimization but is the high-level semantic alignment.
>
> 3. ImageBind typically uses one token to compute the token distances, which is a coarse alignment. ThinkDiff excels at fine-grained alignment for longer token sequences. Even if using more tokens in ImageBind, the lengths of two types of tokens must be the same.
>
> Therefore, as shown in the experiment in **Q1**, our method can achieve much better accuracy compared to the ImageBind-style alignment.

---

### Decision · Program_Chairs · 2025-05-01

**Decision:**

Accept (poster)

**Comment:**

All reviewers found the proposed method effective and the results promising. The rebuttal successfully addressed reviewer comments, and all reviewers recommended acceptance. The authors are encouraged to improve the final version by incorporating reviewer suggestions – particularly by adding further results, providing thorough explanations, and offering in-depth discussions of the method’s advantages and limitations compared to state-of-the-art approaches, as provided in the rebuttal.

Notably, Reviewer j31i, while leaning slightly toward acceptance, raised two remaining concerns to be addressed in the revision: (1) some benchmark methods may not have been specifically trained or fine-tuned on the CoBSAT dataset, potentially making comparisons less fair; and (2) additional evaluation on GenEval and DPGBench suggests that the method’s strength lies more in reasoning than in precise prompt following, as acknowledged by the authors – these should be clearly articulated in the final version.